# CaloFlow for *CaloChallenge* Dataset 1

Claudius Krause[1,2], Ian Pang[1], and David Shih[1]

**1** NHETC, Dept. of Physics and Astronomy, Rutgers University, Piscataway, NJ, USA
**2** Institut für Theoretische Physik, Universität Heidelberg, Germany

May 17, 2024

## Abstract

CaloFlow is a new and promising approach to fast calorimeter simulation based on normalizing flows. Applying CaloFlow to the photon and charged pion Geant4 showers of Dataset 1 of the Fast Calorimeter Simulation Challenge 2022, we show how it can produce high-fidelity samples with a sampling time that is several orders of magnitude faster than Geant4. We demonstrate the fidelity of the samples using calorimeter shower images, histograms of high level features, and aggregate metrics such as a classifier trained to distinguish CaloFlow from Geant4 samples.

# 1  Introduction

The LHC has just restarted for its third run in the spring of 2022, and the need for accurate fast simulation at the LHC is ever more urgent. Simulation of calorimeter showers with GEANT4 [1–3] is already a major computational bottleneck at the LHC, and this is expected to further intensify as the detectors are upgraded and the luminosity is increased [4].

Recently, there have been significant advancements in fast calorimeter simulation through the application of deep generative models such as Generative Adversarial Networks (GANs), normalizing flows, Variational Autoencoders (VAEs) and other approaches [5–20] In CALOFLOW-I [9] and CALOFLOW-II [10], two of us have demonstrated the effectiveness of normalizing flows in emulating GEANT4 events with high-fidelity at generation speeds that are $10^4$ times faster than GEANT4. Also, we found that CALOFLOW is robust against mode collapse that is a common problem in GAN-based generative models. In fact, CALOFLOW proved to be the first ever generative model in HEP that could pass the following stringent test: a binary classifier trained on the raw voxels of (CALOFLOW) generated vs. (GEANT4) reference showers could not distinguish the two with 100% accuracy. Normalizing Flows showed similar good performance in other generative tasks in high-energy physics [21–34].

In this paper, we adapt CALOFLOW to Dataset 1 from the *Fast Calorimeter Simulation Challenge 2022* [35] (hereafter referred to as the *CaloChallenge*). The *CaloChallenge* is aimed at encouraging the development of fast and high-fidelity calorimeter shower generation through the application of deep generative models. It includes three datasets with increasing dimensionality. Dataset 1 [36] has the lowest dimensionality and consists of photon $\gamma$ (368 voxels) and charged pion $\pi^+$ (533 voxels) showers. This dataset is actually the official ATLAS simulation dataset used to develop the FASTCALOGAN model [8] used in a portion of ATLFAST3 [7], which is currently the official fast calorimeter simulation framework of the ATLAS collaboration. It is more realistic than the GEANT4 dataset [37] used in [5, 6, 9, 10] because it includes a realistic sampling fraction and calorimeter geometry.

We will demonstrate the robustness of CALOFLOW even when applied to this new dataset. We will show that CALOFLOW can achieve comparably fast generation times as FASTCALOGAN, with a significant improvement in generation quality. We will also show that CALOFLOW compares favorably (on Dataset 1), in both generation time and quality, to CALOSCORE [12]. This is a recent approach that uses score-based diffusion models and is currently the only approach that has been trained successfully on all three datasets of the *CaloChallenge*.

The outline of the paper is as follows. In Section 2, we provide a brief description of *CaloChallenge* Dataset 1 (for more details, see [35]) and outline the differences compared to the CALOGAN dataset. In Section 3, we elaborate on the method we used to learn the distribution of showers from the GEANT4 reference datasets and the modifications we have made to adapt the previous version of CALOFLOW to *CaloChallenge* Dataset 1. We include the main results of the study in Section 4. Here we show detailed comparison between the CALOFLOW generated samples and the reference GEANT4 samples. Finally, we summarize our findings in Section 5 and include possibilities for future research.

| | Number of voxels | $N_z$ | $N_\alpha$ | $N_r$ |
|---|---|---|---|---|
| $\gamma$ | 368 | 5 | [1, 10, 10, 1, 1] | [8, 16, 19, 5, 5] |
| $\pi^+$ | 533 | 7 | [1, 10, 10, 1, 10, 10, 1] | [8, 10, 10, 5, 15, 16, 10] |

Table 1: Details of $\gamma$ and $\pi^+$ shower datasets: $N_z$ is the total number of calorimeter layers, $N_\alpha$ is the number of $\alpha$ bins in a given layer, and $N_r$ is the number of radial bins in a given layer. $N_\alpha$ and $N_r$ are listed according to increasing layer numbers (i.e. the number of $\alpha/r$ bins in layer 0 are leftmost on the list)

## 2    Dataset

As stated in Section 1, there are three datasets in the *CaloChallenge* and our study focuses only on Dataset 1 [36], as the original version of CaloFlow does not scale well to the dimensionalities of datasets 2 and 3. This dataset comprises calorimeter shower events for $\gamma$ and $\pi^+$ particle types and is a subset of the ATLAS Geant4 datasets that are published at [38]. The datasets described here have been used to train the ATLAS GAN-based model FastCaloGAN [8] used in AtlFast3 [7].

The calorimeter setup is based on a voxelized version [7] of the current ATLAS detector configuration in the $\eta$ range [0.2, 0.25]. As shown in left diagram in Figure 1, the calorimeter layers are simulated as concentric cylinders aligned along the $z$-axis, defined as the direction of travel of the particle. Working in cylindrical coordinates, $\Delta\phi$ and $\Delta\eta$ are defined as the $x$ and $y$-axes respectively. Based on the voxelization, each calorimeter is further divided into radial and angular bins as shown in the right diagram in Figure 1. Here $r \equiv \sqrt{(\Delta\phi)^2 + (\Delta\eta)^2}$ and $\alpha \equiv \arctan\left(\frac{\Delta\eta}{\Delta\phi}\right)$. The total number of voxels, and the arrangement of voxels in $z$, $\alpha$ and $r$ for the $\gamma/\pi^+$ datasets are shown in Table 1. For comparison, the CaloGAN calorimeter setup consists of 3 calorimeter layers and a total of 504 voxels regardless of the particle type.

The incident energies are discrete and range from $2^8$ MeV to $2^{22}$ MeV, increasing in powers of 2. Hence, there are 15 different incident energies in total. (In contrast, the incident energies in the CaloGAN dataset were uniformly sampled.) The photon and pion datasets each contain 10000 events for every low incident energy. Fewer events were generated for each high incident energy due to the longer generation time with Geant4. Figure 2 shows the distribution of incident energies in the photon and pion datasets.

In Dataset 1, there are two independent photon datasets with 121000 events each, and two independent pion datasets with 120800 events each. For both particle types, the first dataset is used for training the generative models (with a 70/30 train/val split); the second is used for training the classifier metric (with a 60/20/20 train/val/test split) and for producing all the evaluation plots. This is ideal, since then any overfitting of the density estimator to the training set could in principle be diagnosed by the independent dataset used for the classifier metric and evaluation plots.

## 3    Method

Our approach follows the algorithm of CaloFlow [9,10] to a large extent. Here we briefly describe the approach, focusing primarily on the differences with [9,10].

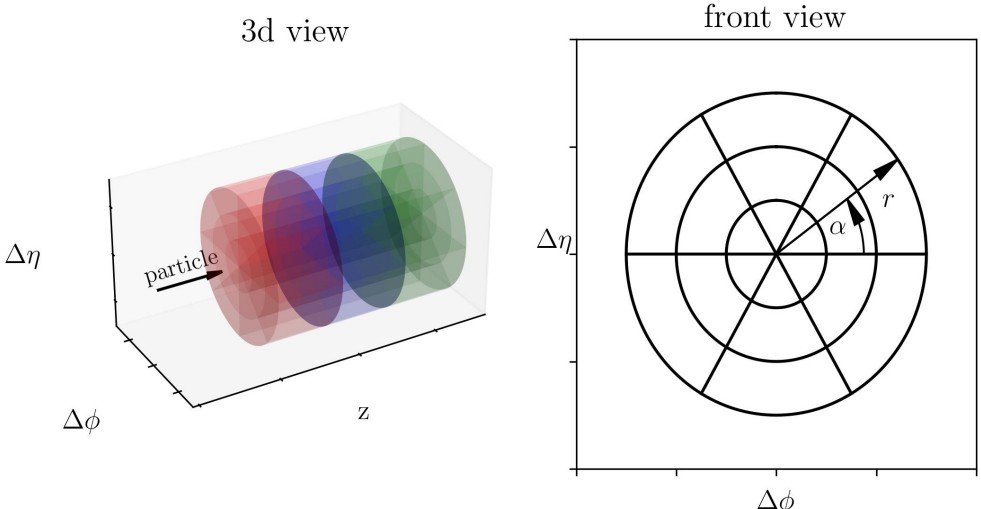

Figure 1: Diagram of coordinate system used in the *CaloChallenge* dataset [35].

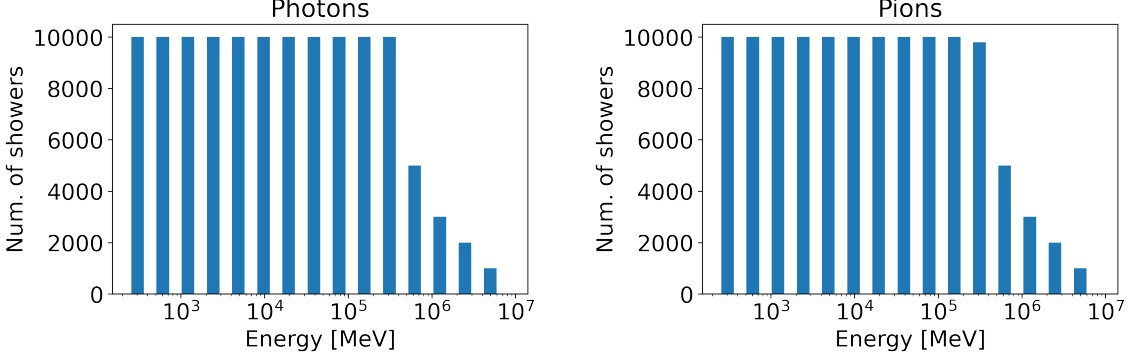

Figure 2: Breakdown of incident energies in *CaloChallenge* Dataset 1.

## 3.1 Normalizing Flows

Normalizing Flows (NFs) (see [39–41] for reviews on NFs) are a class of machine learning models for density estimation and generative modeling that aim to learn a bijective transformation (with tractable Jacobian) between data and a latent space following a simple base distribution (e.g. the normal distribution). Since the probability of a point of the base distribution, as well as the Jacobian of the transformation is known, NFs can give the probability density $p(x)$ of each data point $x$ in data space, i.e. NFs are density estimators. When run in the other direction, starting from samples $z$ in the latent space generated from the base distribution, NFs can be used as generative models. Since by construction, the log-likelihood (LL) of datapoints under the flow are known, a NF can be trained by minimizing the negative LL.

Following [9, 10], the NFs used here are based on the Masked Autoregressive Flow (MAF) [42] and Inverse Autoregressive Flow (IAF) [43] architectures with rational quadratic spline (RQS) transformations [44, 45]. The autoregressive parameters of the RQS transformations are determined using neural networks known as MADE [46] blocks. This maximizes the expressivity of the NF, but comes at the expense of unbalanced evaluation times in the forward and inverse direction. MAFs are fast for density estimation but slow for sampling, while IAFs are fast for sampling but slow for density estimation. As in [9,10],

we will refer to the MAF as the "teacher" model and the IAF as the "student" model, as the MAF was used as an intermediate step to obtain the trained IAF model, which is final product of the CaloFlow method.

MAFs can be trained efficiently with the LL objective described above, but for IAFs this is usually not possible due to time and memory constraints. Training the IAF efficiently requires a different approach known as *Probability Density Distillation* (PDD) [47] or *teacher-student training*. Instead of fitting the IAF directly to the data, the approach involves fitting the IAF to the MAF. In practice, the fitting is implemented based on two training loss terms that we refer to as $z$ and $x$-losses. To compute the $z$-loss, we begin with a sample $z$ which is then passed through the student IAF to obtain a sample $x'$ in data space and the corresponding likelihood $s(x')$. The data sample $x'$ is then mapped via the teacher MAF back to the latent space which obtains the likelihood $t(x')$. Similarly, to compute the $x$-loss, one can start with a data sample $x$ which maps to latent space $z'$ via the teacher, and then map back to data space via the student. In the original PDD study [47] study, the KL divergence of $s(x')$ and $t(x')$ was initially used as the training loss. However, the authors noted that it does not converge well. Hence, as in [10], we used a training loss function that is based on a mean square error that compares relevant values[1] at each equivalent stage of the teacher and student passes. The total loss function is then the sum of the $x$ and $z$-losses. Such a loss function has proven effective in matching the student model to the teacher model. For more details we refer to [10].

## 3.2   CaloFlow

CaloFlow [9,10] learns the distribution of calorimeter showers in voxel space conditioned on the incident energy, $p(\vec{\mathcal{I}}|E_{\text{inc}})$, in a two-step setup. In the first step, a normalizing flow called flow-I learns the distribution of energy depositions in the layers of the calorimeter $E_i$ conditioned on the incident energy, $p_1(E_i|E_{\text{inc}})$. A second normalizing flow, called flow-II, learns the normalized showers conditioned on these energies, $p_2(\vec{\mathcal{I}}|E_i, E_{\text{inc}})$. Normalized in this context means that the sum of energy depositions in all voxels per calorimeter layer is 1.

When adapting CaloFlow to *CaloChallenge* Dataset 1, we retained most of its key features, while making several important modifications that are elaborated on below:

1. **Preprocessing:**

   (a) **Unit-space definition:**
       In [9,10], the energy deposition in the calorimeter layers were recursively transformed to $\vec{u} \in [0,1]^{N_z}$, where $u_0$ (the first element in $\vec{u}$) was defined by:

       $$u_0 = \frac{\sum_{i=0}^{N_z-1} E_i}{E_{\text{inc}}}, \tag{1}$$

       In this study, we had to modify $u_0$ to account for cases where $\sum_{i=0}^{N_z-1} E_i > E_{\text{inc}}$:[2] we rescaled $u_0$ by its maximum value taken over all showers in the training set.[3]

---

[1]For $\gamma$ student, these consist of coordinates before and after passing them through the flows and RQS parameters from individual MADE blocks within the bijectors. For $\pi^+$ student, we did not enforce agreement with the teacher at the level of individual MADE blocks, but only at the endpoints of the flows.

[2]Naively, we might think that $E_{\text{inc}} \geq \sum_{i=0}^{N_z-1} E_i$ due to the conservation of energy. However, this is not always true due to rescaling that is done on the true deposited energy to account for the sampling fraction ($< 1$) of the calorimeter when creating the training/evaluation datasets.

[3]There was an exception when training on the $\gamma$ dataset. We found that the event with the largest ratio of total deposited energy to incident energy was an outlier ($u_0 = 3.04$) and hence decided to ignore that single event when training our model.

Specifically, we rescaled by $\max(u_0) = 2.42$ for $\gamma$ and $\max(u_0) = 6.93$ for $\pi^+$. This rescaling of $u_0$ ensures that $u_0 \in [0, 1]$ for the showers of Dataset 1.

(b) **Flow-I noise level for $\pi^+$:**

For the flow-I of $\pi^+$ dataset, we used a smaller range for the uniform random noise $[0, 0.1]$ keV that is added to the voxel energies prior to summing them to get the layer energies. We observe that a lower noise level improved the training of $\pi^+$ flow-I. We kept the original noise range $[0, 1]$ keV when training flow-I on the $\gamma$ dataset and flow-II for both particle types as lowering the noise level worsened the classifier score.

(c) **Preprocessing of incident energies:**

The incident energy $E_{\text{inc}}$ is fed to flow-I and flow-II as a conditional label in the form:
$$\log_{10}\left(E_{\text{inc}}/33.3 \text{ GeV}\right) \in [-2.5, 2.5] \tag{2}$$

(In [9, 10], 10 GeV was used as the normalization point.)

(d) **Preprocessing of layer energies (Flow-II):** The preprocessing of layer energies used as conditional labels in flow-II is:

$$\log_{10}\left((E_i + 1 \text{ keV})/100 \text{ GeV}\right) - 1 \in [-2, 4] \tag{3}$$

which is slightly different than that of [9, 10].

2. **Number of training epochs:** See Table 2 for changes in the number of training epochs.

| | | Number of training epochs (Training time) | |
|---|---|---|---|
| | | Teacher | Student |
| $\gamma$ | flow-I | 100 (46 min) | - |
| | flow-II | 100 (77 min) | 100 (360 min) |
| $\pi^+$ | flow-I | 100 (52 min) | - |
| | flow-II | 100 (98 min) | 150 (520 min) |

Table 2: Number of training epochs used in CALOFLOW when training on *CaloChallenge* Dataset 1. The corresponding training times, obtained on our TITAN V GPU, are included in parentheses. Note that flow-I is only trained once per dataset ($\gamma$ or $\pi^+$), and then used for both the teacher and the student models.

3. **Hyperparameters:**

(a) **Hidden layer/Batch sizes:**

We experimented with different hidden layer sizes when adapting CALOFLOW to *CaloChallenge* Dataset 1. At times, larger hidden layer size leads to lower losses. However, at other times, it was surprising that the performance went down and the loss went up during training; this could be a sign of overfitting. Memory size was also a limitation when deciding on hidden layer size. Finally, we settled with the following hidden layer sizes: 378 for $\gamma$ teacher; 736 for $\gamma$ student;

533 for $\pi^+$ teacher; 500 for $\pi^+$ student. A summary of the MADE [46] block architecture features for the various models are shown in Table 3.

When training flow-I, we used a batch size of 200 for both particle types. When training flow-II, we used batch sizes of 500 for $\pi^+$ teacher and 200 for $\gamma$ teacher. For the student models, we used the batch size of 175 as stated in [9, 10]. This choice of hyperparameters enabled us to achieve better overall classifier scores.

|  |  | input | hidden | output |
|---|---|---|---|---|
| $\gamma$ | Teacher | 378 | $1 \times 378$ | 8464 |
|  | Student | 736 | $1 \times 736$ | 8464 |
| $\pi^+$ | Teacher | 533 | $1 \times 533$ | 12259 |
|  | Student | 500 | $1 \times 500$ | 12259 |

Table 3: Summary of MADE block architecture of the various teacher and student models. The number nodes in the input layer, hidden layer and output layer are shown for each of the 4 models.

(b) **$E_{\mathbf{inc}}$ distribution in $z$-loss:**

Previously in [9,10], we used the same uniform $E_{\mathrm{inc}}$ distribution as the training data when feeding the conditional label through the student model to compute the $z$-loss. In general, it does not have to be the case. In this study, we have a discrete $E_{\mathrm{inc}}$ distribution in the training data. We found that using this same distribution works well when computing the $z$-loss for the $\gamma$ student model. In contrast, we observed that using a uniform $E_{\mathrm{inc}}$ distribution that has the same range as the training data works better when computing the $z$-loss for the $\pi^+$ student model.

4. **Learning rate (LR) schedule:**

Instead of the simple multi-step decreasing LR schedule used in [9, 10], here we adopted a cyclic LR schedule [48] (see Appendix A for details) when training flow-I and flow-II, as this was found to result in significantly improved performance. The effectiveness of cyclic LR schedule may be due the use of higher LRs at certain points in the training which helps the model move out of saddle points or local minima. One key difference in the implementation of cyclic LR is that the LR is updated after each batch in contrast to updating after each epoch milestone in [9, 10]. As in [49], we observe that the OneCycle LR policy — a special case of cyclic LR which only relies on a single cycle — generally allows us to shorten the training time by obtaining a lower loss with a smaller number of training epochs. This is especially useful when training the student model (IAF) which generally has a longer training time per epoch compared to training the teacher model (MAF).

When experimenting with LR schedules, we found that training with OneCycle LR often achieved better results compared to regular cyclic LR, so we adopted this throughout. The only exception was for $\gamma$ teacher where training with regular cyclic LR had a better outcome.

# 4 Results

## 4.1 Average shower images

In Figures 3 and 4, we compare the average shower images between the samples generated by CALOFLOW and GEANT4 for the $\gamma$ and $\pi^+$ datasets respectively. Looking at the average shower images for both samples, we see that they are virtually indistinguishable by eye. A more detailed comparison between the CALOFLOW and GEANT4 samples can be seen by looking at the histograms in Section 4.2.

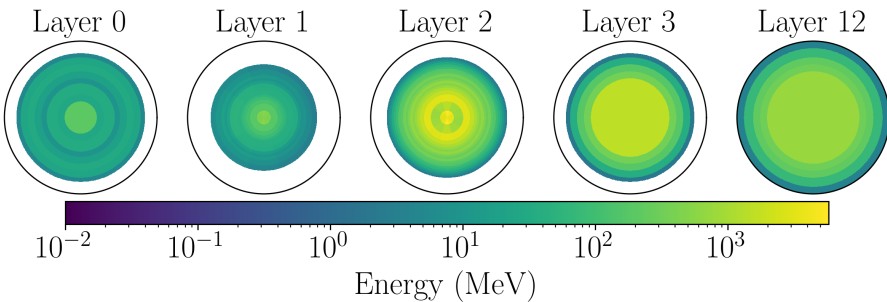

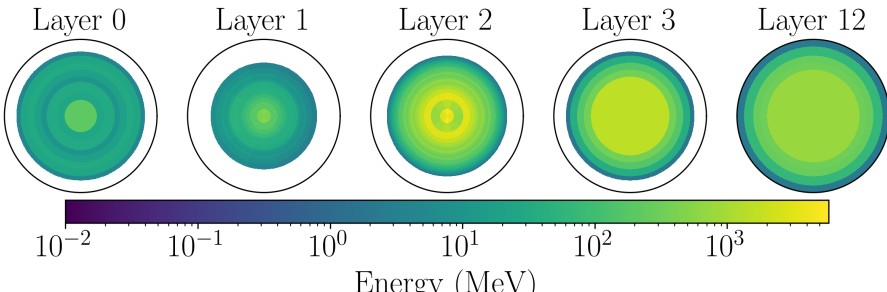

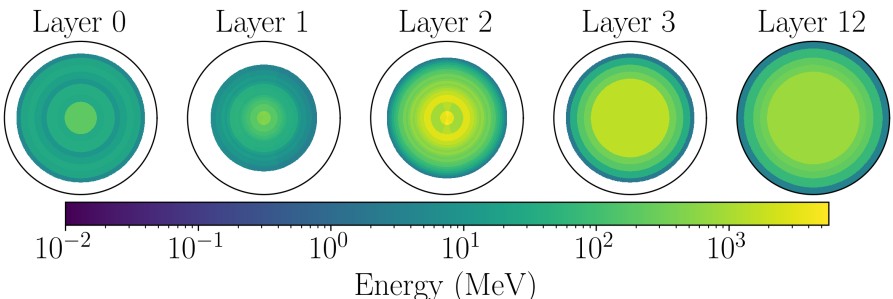

Figure 3: Shower averages for $\gamma$ teacher (top), $\gamma$ student (middle) and GEANT4 $\gamma$ reference dataset (bottom) respectively.

Shower average pion teacher

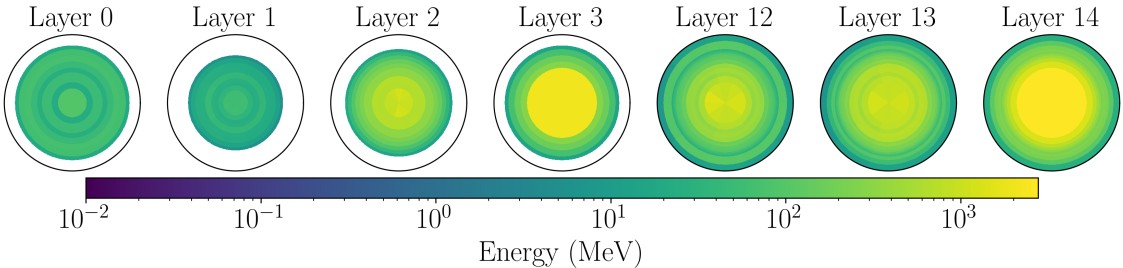

Shower average pion student

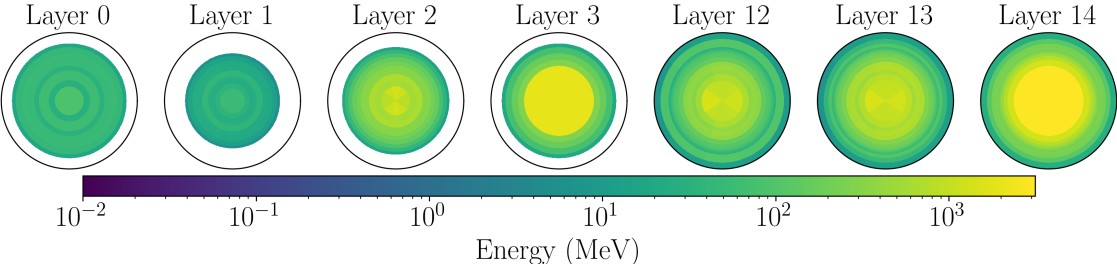

Shower average GEANT4 pion reference dataset

Figure 4: Shower averages for $\pi^+$ teacher (top), $\pi^+$ student (middle) and GEANT4 $\pi^+$ reference dataset (bottom) respectively.

## 4.2 Histograms

### 4.2.1 Energy histograms

Figures 5–8 show histograms corresponding to energy distributions produced by CALOFLOW compared to those from the GEANT4 reference sample. In Figure 5, we see that, despite having more calorimeter layers than before, flow-I is still able to precisely learn the $\gamma$ layer energies. Also, the transformation to unit-space in the preprocessing ensured that $E_{\text{tot}}/E_{\text{inc}}$ is learned well and this is clearly reflected in the $E_{\text{tot}}/E_{\text{inc}}$ histogram at the bottom right of Figure 5. We are pleasantly surprised at how well flow-I was able to learn the spikes found at higher energies in the $E_2$ plot. There are some minor discrepancies at higher energies which can be attributed to there being fewer events with high $E_{\text{inc}}$ as shown in Figure 2.

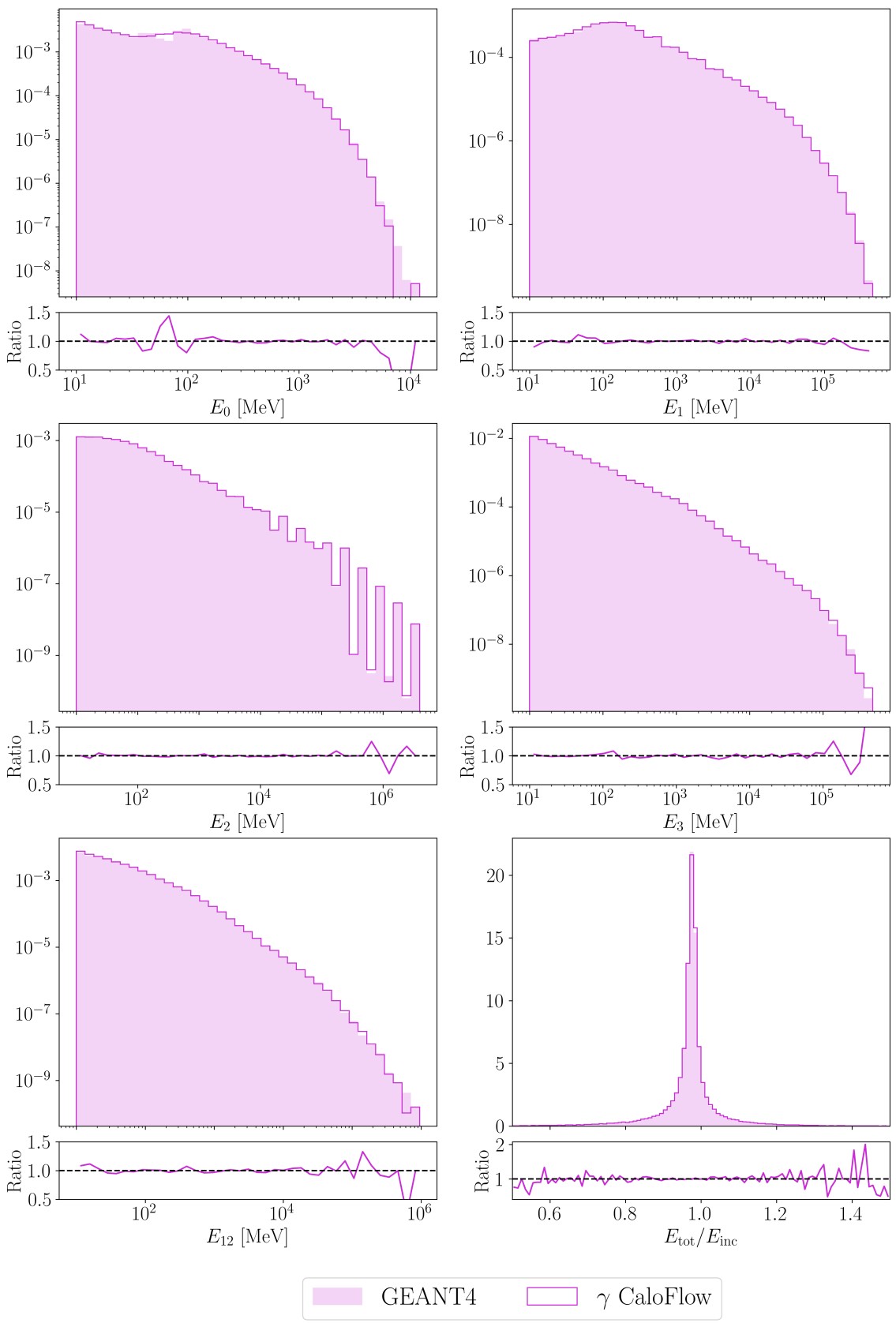

Figure 5: Energy distributions for $\gamma$ dataset.

Similarly, we see from Figure 6 that flow-I was able to learn the $\pi^+$ layer energies very

accurately. Initially, we found that flow-I struggled with learning the bimodal distribution found in the first 3 layer energy histograms (top row in Figure 6). However, training flow-I with a lower noise level helped it to better learn complicated distributions in the $\pi^+$ dataset. Like for the $\gamma$ dataset, $E_{\text{tot}}/E_{\text{inc}}$ is well-learned by flow-I. The slight discrepancy at high energies can be attributed to there being fewer events with high $E_{\text{inc}}$ as shown in Figure 2.

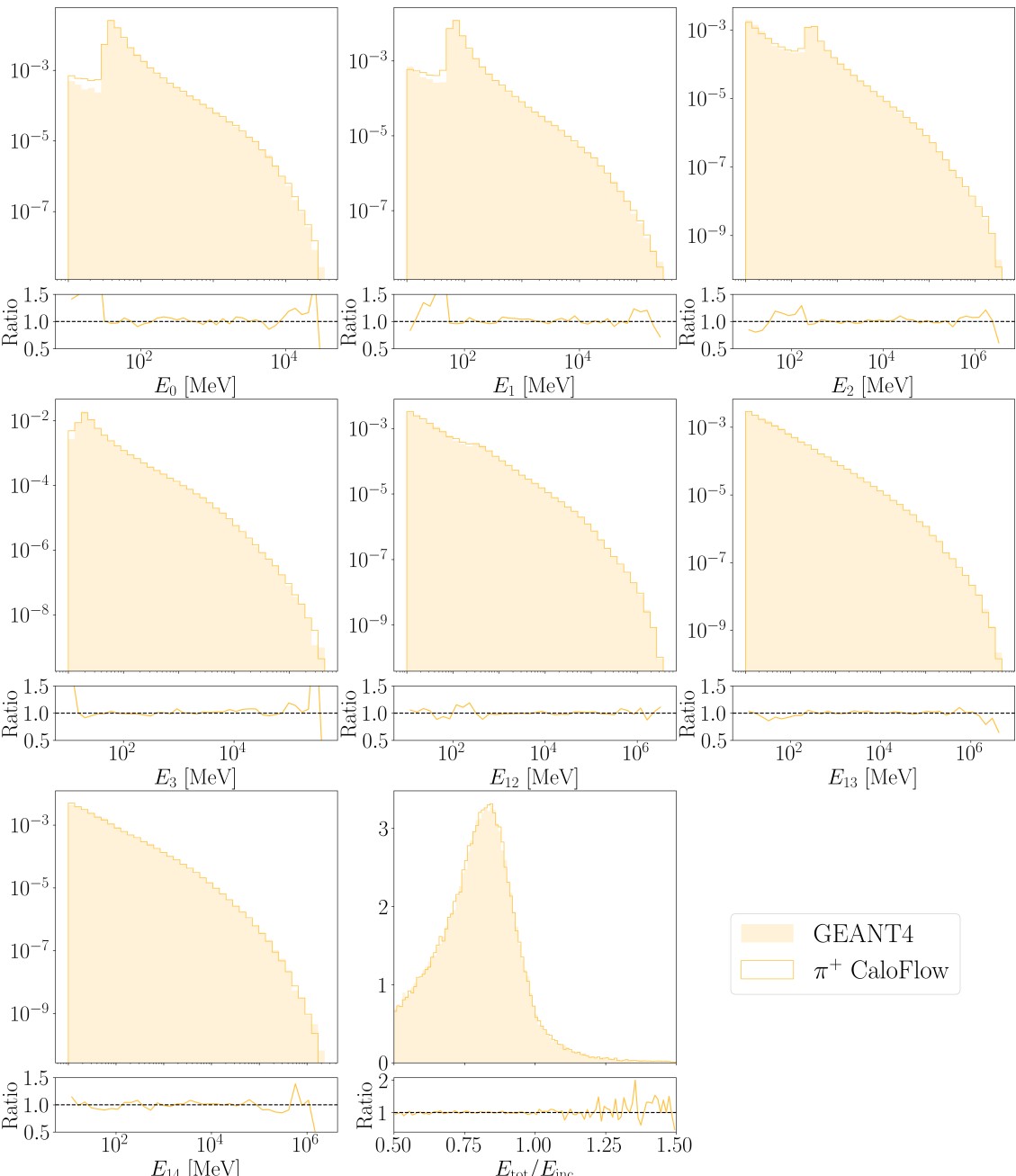

Figure 6: Energy distributions for $\pi^+$ dataset.

Next, we compare the distributions of $E_{\text{tot}}/E_{\text{inc}}$ from the CALOFLOW and GEANT4 samples for various discrete incident energies $E_{\text{inc}}$. The corresponding histograms are shown in Figures 7 and 8. An equivalent comparison can be found for the ATLFAST3 study in Figures 10 and 11 of [7]. As additional comparison, we included a digitized [50] version of the

ATLFAST3 [7] result. Overall, we find that CALOFLOW is able to properly learn the mean and widths of the $E_{\text{tot}}/E_{\text{inc}}$ distributions. CALOFLOW is even able to reproduce most of the distributions corresponding to higher $E_{\text{inc}}$ which have lower statistics. By using lower noise level when training $\pi^+$ flow-I, we were able to properly describe the bimodal distribution found in the $\pi^+$ histograms corresponding to 256 MeV, 512 MeV, and 1024 MeV. This is a significant improvement from the results found in the ATLFAST3 study.

In Figures 9 and 10, we include plots that show a detailed comparison between the histograms generated based on CALOFLOW and GEANT4 shown in Figures 7 and 8. Following FASTCALOGAN [51], we calculated the mean and RMS from the histograms which have excluded outliers. In Figure 9, we find that there is less than 0.3% discrepancy in the mean value of each $\gamma$ histogram in Figure 7. There is a larger discrepancy in the RMS values. Nevertheless, the maximum discrepancy in RMS is still only about 5% and it occurs at high $E_{\text{inc}} = 2.1$ TeV which has lower statistics. Most of the RMS discrepancies for lower $E_{\text{inc}}$ are less than 2%. In Figure 10, we find that all the $\pi^+$ histogram means in Figure 8 have less than 2% discrepancy. We also see excellent agreement in the RMS with less than 3% discrepancy for most of the histograms. Comparing with the equivalent results from ATLFAST3 (see Figures 12(a) and 12(c) in [7]), we see that CALOFLOW managed to achieve a greater degree of agreement with GEANT4.

As an additional comparison with the ATLFAST3 results, we computed the $\chi^2$ per degree of freedom ($\chi^2$/NDF) for the histograms in Figures 7 and 8. Although we used a slightly different binning compared to ATLFAST3, dividing $\chi^2$ by NDF should make the calculated values robust against different binning choices. The results can be found in Table 4. The ATLFAST3 result was taken from [7] and not from our digitization, as the line extraction is not very accurate. We find that the $\chi^2$/NDF values of the CALOFLOW results in Figures 7 and 8 are $\mathcal{O}(10)$ better than the corresponding results found in ATLFAST3, indicating again that normalizing flows generate higher quality showers compared to GANs [9, 10].

| | $\chi^2$/NDF | |
| --- | --- | --- |
| | CALOFLOW (this work) | ATLFAST3 [7] |
| $\gamma$ | $526/450 = \mathbf{1.17}$ | $5657/419 = 13.5$ |
| $\pi^+$ | $629/480 = \mathbf{1.31}$ | $5503/435 = 12.7$ |

Table 4: Comparison of $\chi^2$/NDF values between CALOFLOW and ATLFAST3 for histograms of $E_{\text{tot}}/E_{\text{inc}}$ at discrete values of $E_{\text{inc}}$ (see Figures 7 and 8).

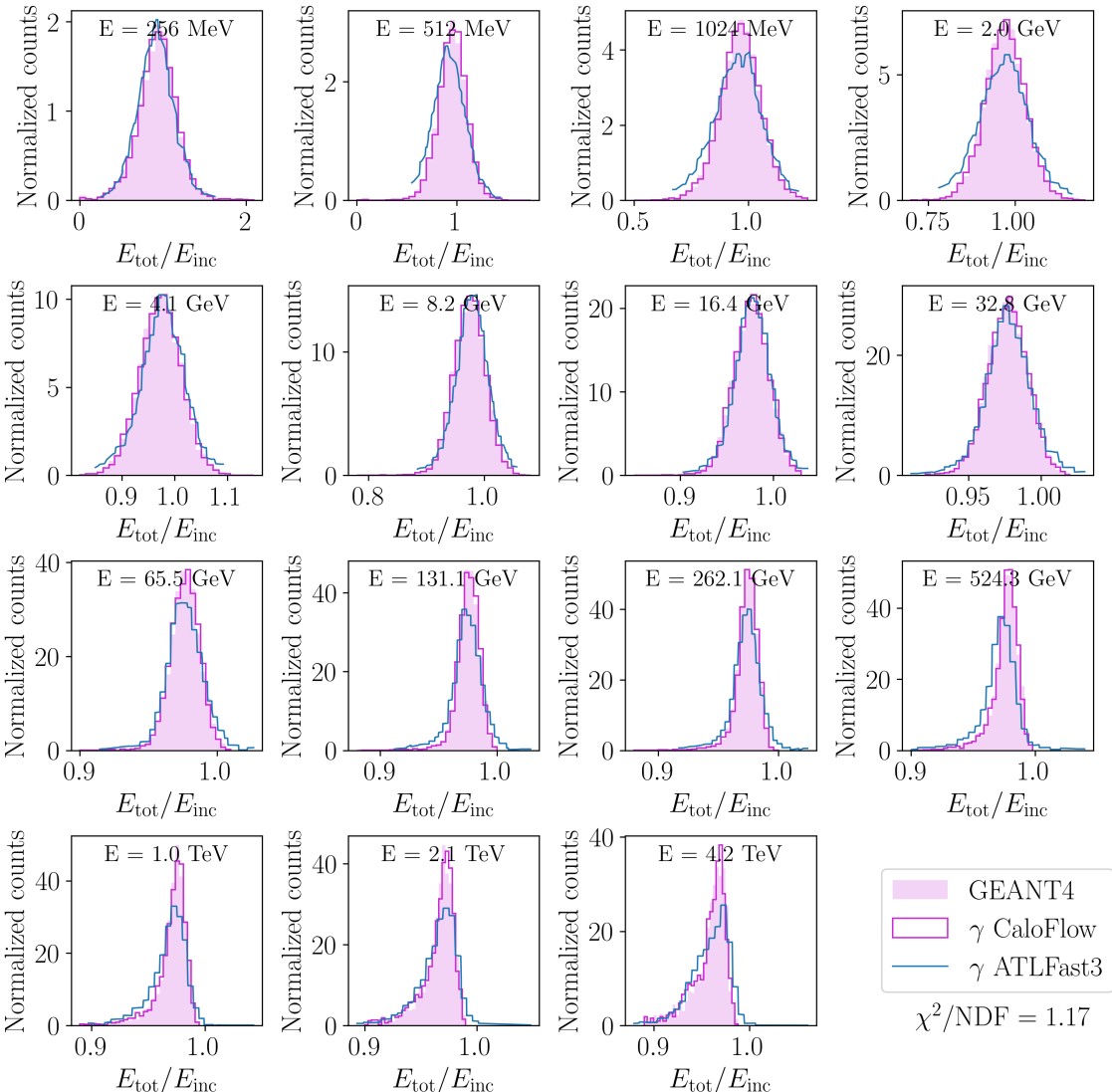

Figure 7: Histograms of $E_{\text{tot}}/E_{\text{inc}}$ for various discrete values of $E_{\text{inc}}$ in the $\gamma$ dataset. To guide the eye, we digitized [50] Figure 10 of AtlFast3 [7].

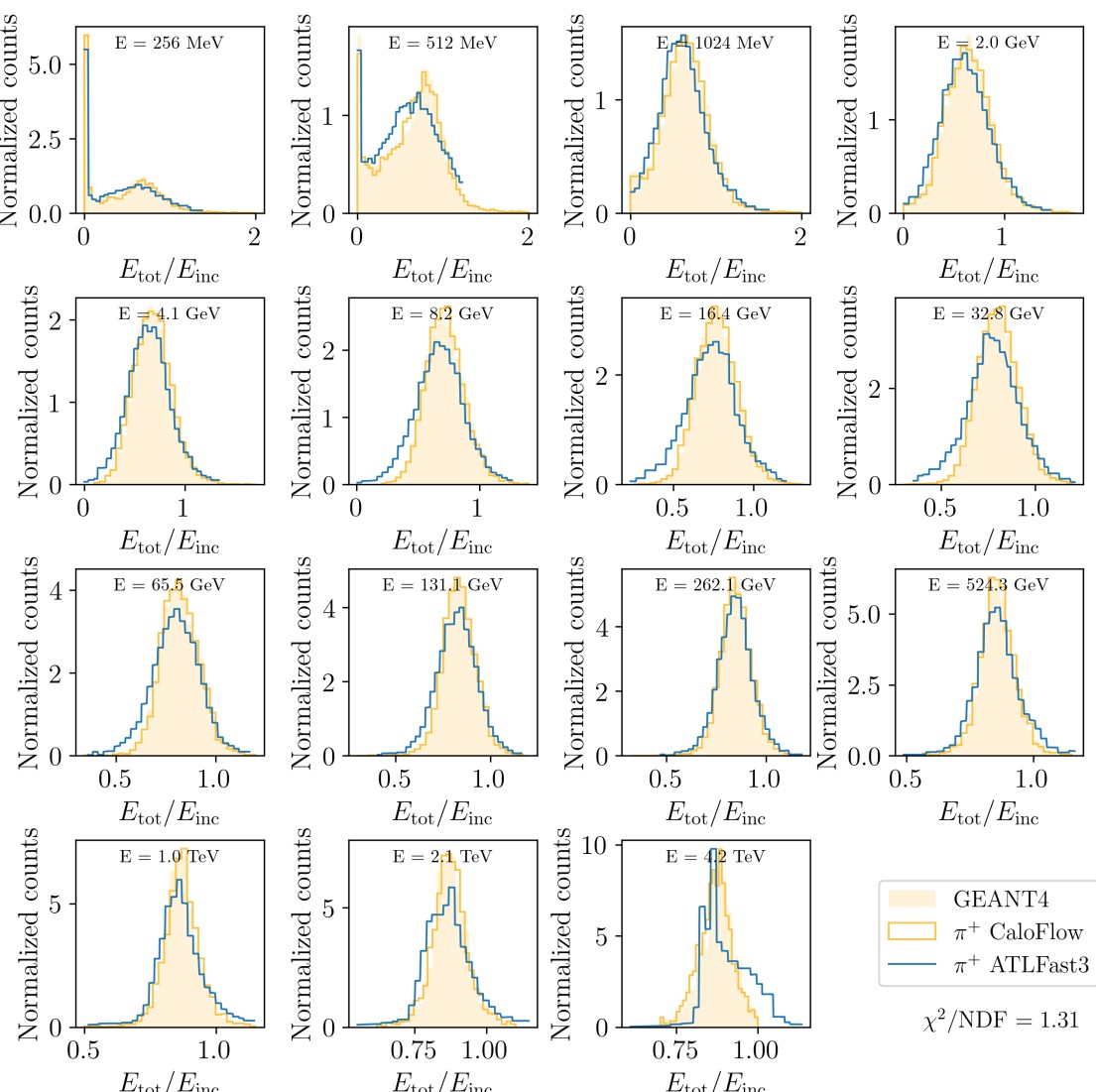

Figure 8: Histograms of $E_{\text{tot}}/E_{\text{inc}}$ for various discrete values of $E_{\text{inc}}$ in the $\pi^+$ dataset. To guide the eye, we digitized [50] Figure 11 of ATLFAST3 [7].

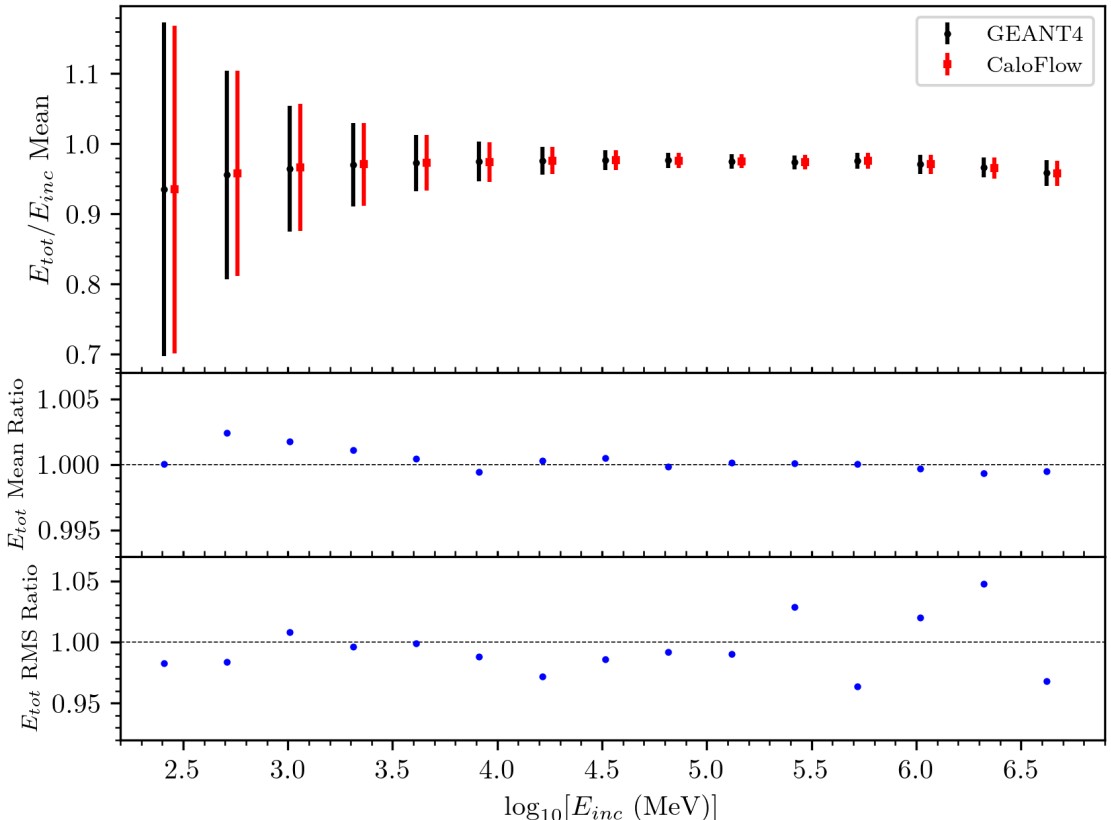

Figure 9: Comparison of mean and RMS of $\gamma$ histograms between CaloFlow and Geant4 in Figure 7. Following FastCaloGAN [51], we calculated the mean and RMS from the histograms which have excluded outliers. Top: Mean of $E_{tot}/E_{inc}$ vs $E_{inc}$. Middle: Ratio of $E_{tot}$ mean between CaloFlow and Geant4. Bottom: Ratio of $E_{tot}$ RMS between CaloFlow and Geant4. The low statistical uncertainty in the data points of the lower two panels resulted in the error bars being smaller than the size of the markers. See Figure 12(a) in [7] for comparison with AtlFast3 study.

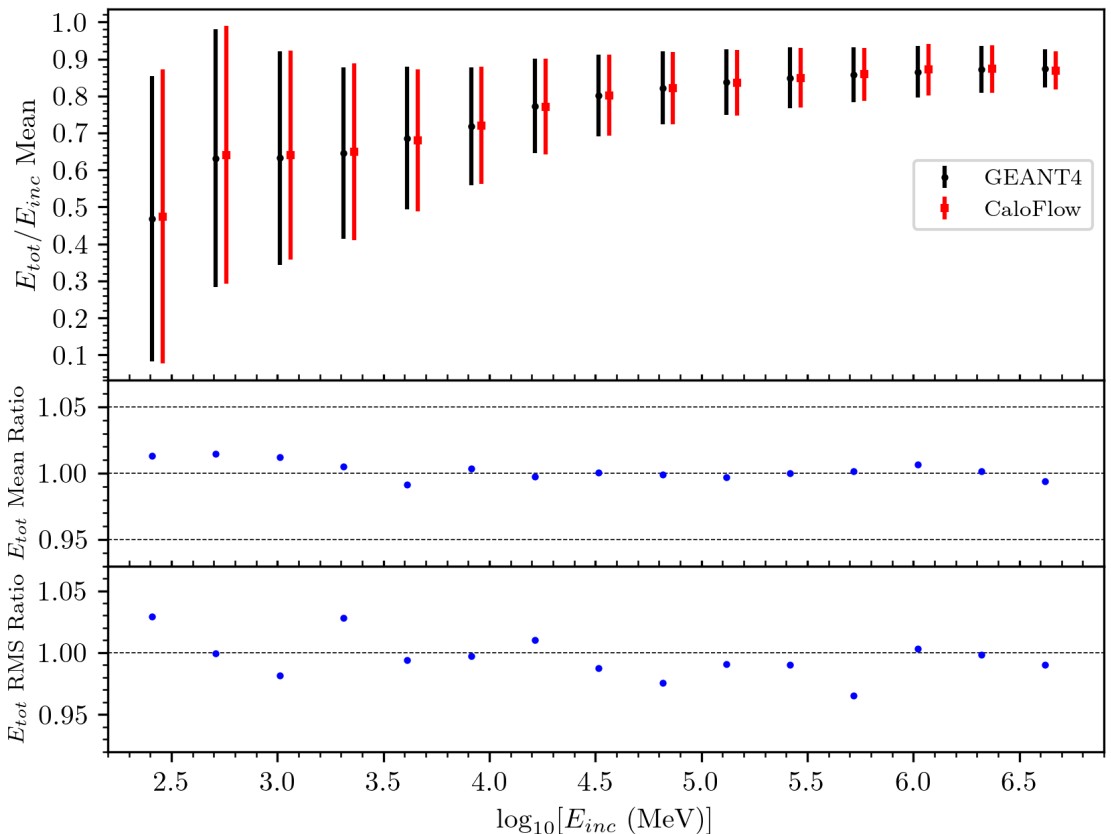

Figure 10: Same as Fig. 9 but for $\pi^+$ showers. See Figure 12(c) in [7] for comparison with ATLFAST3 study.

### 4.2.2 Shower shape histograms

Figures 11–16 show histograms corresponding to shower shape distributions produced by CALOFLOW compared to those from the GEANT4 reference sample. The center of energy in the $\eta$ and $\phi$ directions are defined based on the location of the voxel centers. The center of energy in the $\eta(\phi)$ direction is defined via the locations of the voxel centers in mm, $H(F)$, as shown in eq. (4)

$$\langle \eta_k \rangle = \frac{\sum \left( \vec{\mathcal{I}}_k \odot H \right)}{E_k} \quad \text{and} \quad \langle \phi_k \rangle = \frac{\sum \left( \vec{\mathcal{I}}_k \odot F \right)}{E_k} \tag{4}$$

where $\vec{\mathcal{I}}_k$ contains the voxel energies in the $k$th layer, $\odot$ denotes the Hadamard product, and $\sum$ denotes sum over all elements.

The corresponding width of the center of energy is defined as shown in eq. (5).

$$\sigma_k^\eta = \sqrt{\frac{\sum \left( \vec{\mathcal{I}}_k \odot H^2 \right)}{E_k} - \left( \frac{\sum \left( \vec{\mathcal{I}}_k \odot H \right)}{E_k} \right)^2} \quad \text{and} \quad \sigma_k^\phi = \sqrt{\frac{\sum \left( \vec{\mathcal{I}}_k \odot F^2 \right)}{E_k} - \left( \frac{\sum \left( \vec{\mathcal{I}}_k \odot F \right)}{E_k} \right)^2} \tag{5}$$

We see in Figures 11–16 that the center of energy in both the $\eta$ and $\phi$ directions modelled by CALOFLOW closely match the ones found in the GEANT4 reference sample. Furthermore, teacher and student histograms are mostly overlapped. This implies that

the CALOFLOW student was very well trained on the CALOFLOW teacher. Due to the close similarity between the teacher and student histogram results, it will not be necessary for us to distinguish between the teacher and student results in the rest of Section 4.2. This result is remarkable considering the fact that there are spikes in the some of the center of energy histograms (e.g. see Fig 14) that might make them difficult to learn. As for the widths of the center of energy, we find two peaks in the distribution for both $\gamma$ and $\pi^+$. For $\gamma$, the global maximum is found away from zero, while the global maximum is found at zero for $\pi^+$. There is some disagreement at larger widths found in Figures 13-16. Still, we find excellent overall agreement in the bimodal distribution of the center of energy widths. Even at larger widths, CALOFLOW is able to model the general shape of the distribution.

Finally, looking at the distribution of voxel energies for all layers in Fig. 17, we see that they are closely modelled by CALOFLOW over 6 orders of magnitude for both, $\gamma$ and $\pi^+$ showers. There is some discrepancy in the low voxel energy region between $[10^{-2}, 10^{-1}]$ MeV especially for the pion showers. There is a peak around $10^{-1}$ MeV corresponding to minimum ionizing particles (MIPs) in the pion voxel energy distribution. However, CALOFLOW is unable to properly model this feature.

Fig. 18 includes 3 voxel energy distributions with each corresponding to a different $E_{\text{inc}}$. Showers with $E_{\text{inc}} = 512$ MeV are taken to be representative of low energy events, while showers with $E_{\text{inc}} = 32768$ MeV are representative of mid energy events, and showers with $E_{\text{inc}} = 2097152$ MeV are representative of high energy events. We found that the MIP peak is present for low and mid $E_{\text{inc}}$ events as shown in Fig. 18. The inability to model the MIP peak might be the reason for the poor fit at low voxel energies for the low and mid $E_{\text{inc}}$ plots. In Section 4.3, we show that a trained binary classifier can be sensitive to the poor fit at low voxel energies.

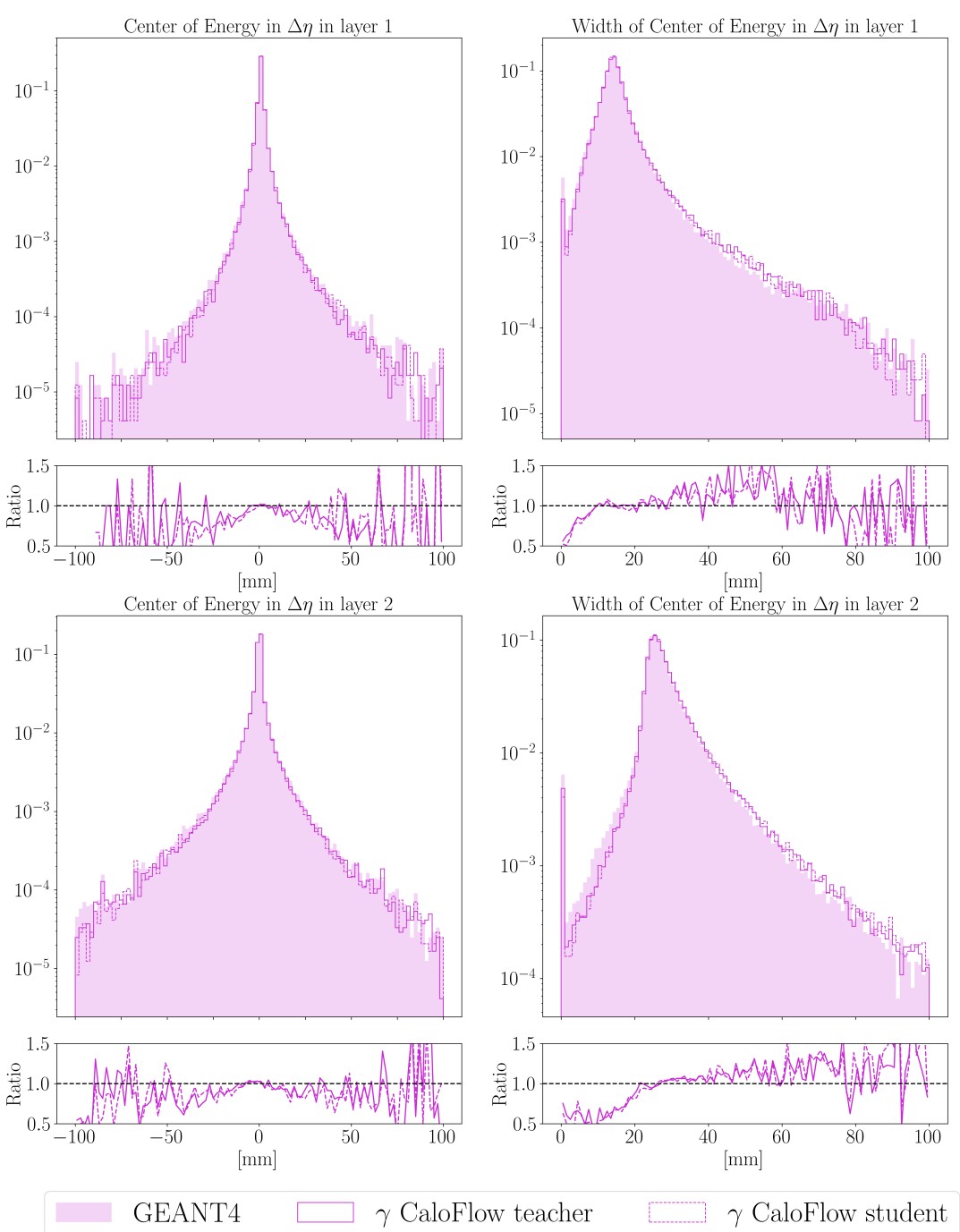

Figure 11: Distributions of the center of energy in the $\eta$ and direction for the $\gamma$ dataset. Note that layers 0, 3 and 12 only have a single $\alpha$ bin each. Hence, there is no positional information to extract for these layers.

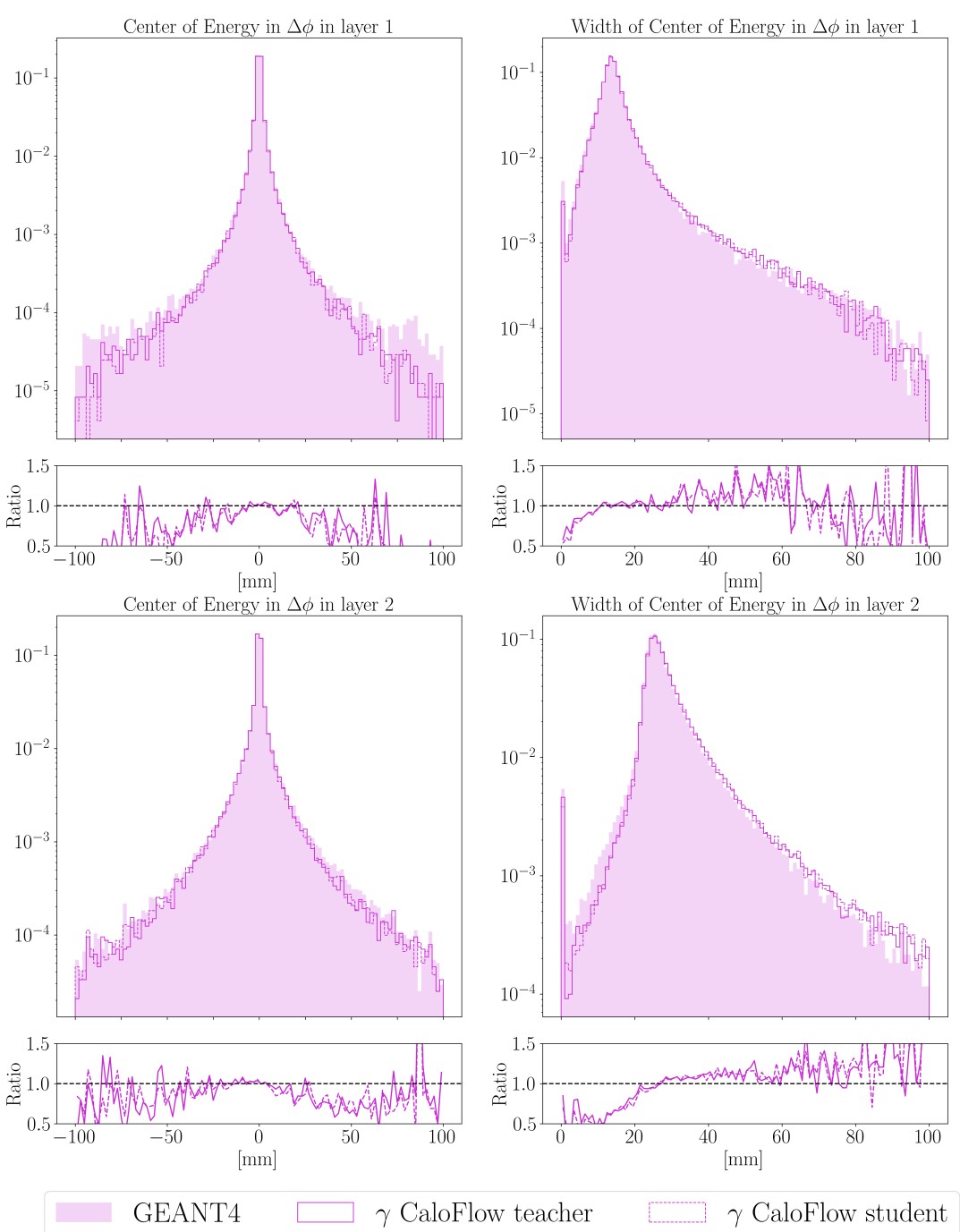

Figure 12: Distributions of the center of energy in the $\phi$ and direction for the $\gamma$ dataset. Note that layers 0, 3 and 12 only have a single $\alpha$ bin each. Hence, there is no positional information to extract for these layers.

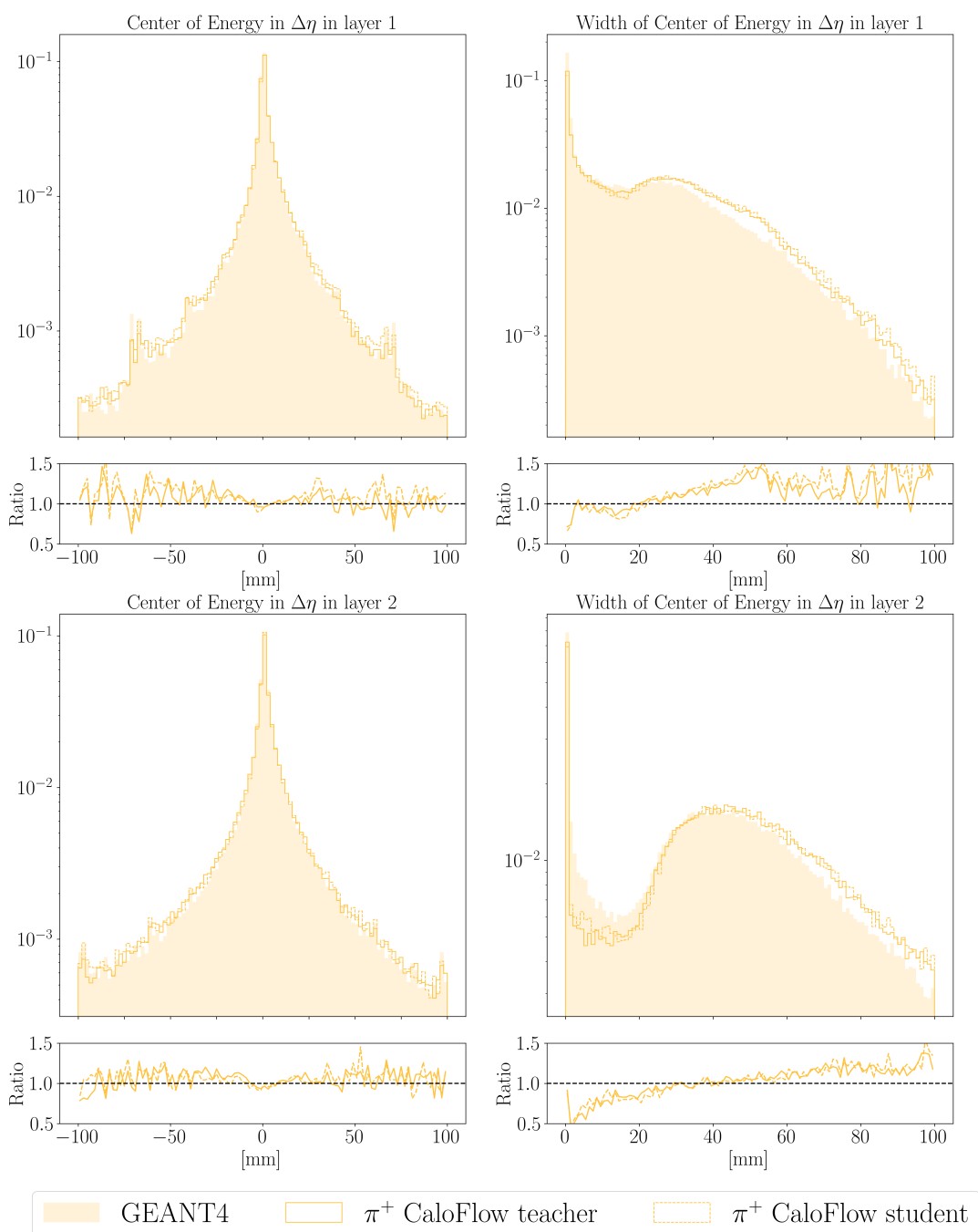

Figure 13: Distributions of the center of energy for layers 1 and 2 in the $\eta$ direction for the $\pi^+$ dataset. Note that layers 0, 3 and 14 only have a single $\alpha$ bin each. Hence, there is no positional information to extract for these layers.

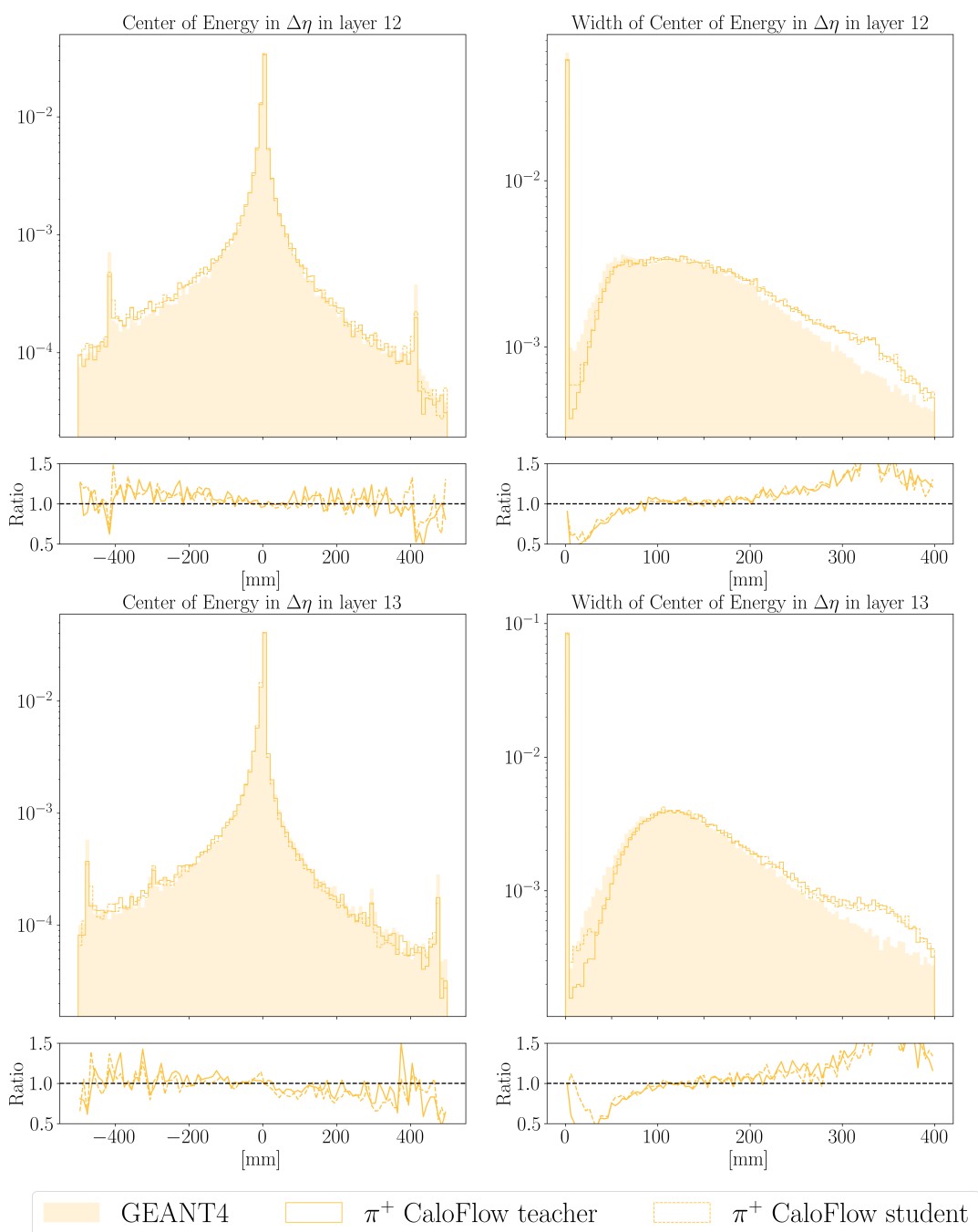

Figure 14: Distributions of the center of energy for layers 12 and 13 in the $\eta$ direction for the $\pi^+$ dataset. Note that layers 0, 3 and 14 only have a single $\alpha$ bin each. Hence, there is no positional information to extract for these layers.

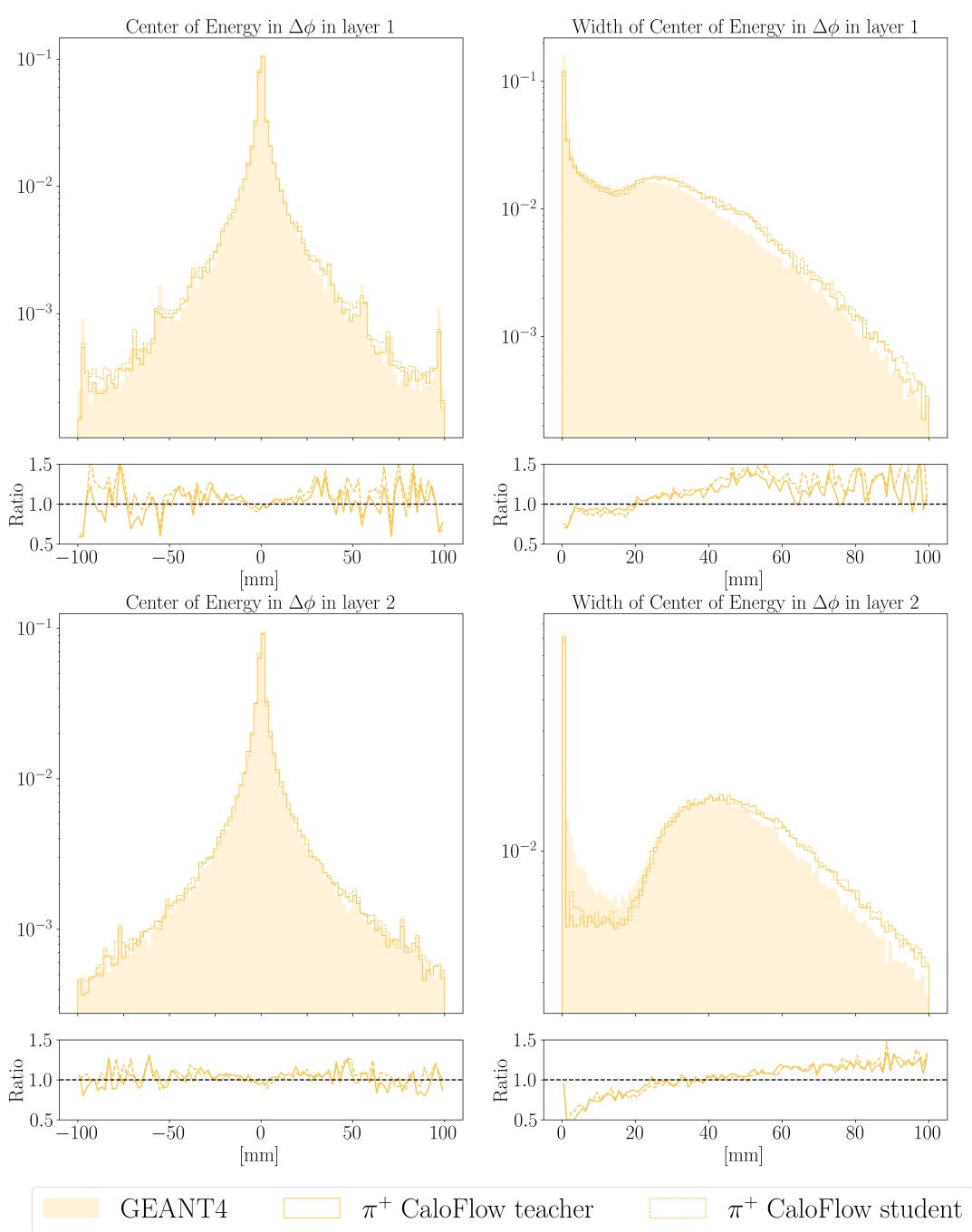

Figure 15: Distributions of the center of energy for layers 1 and 2 in the $\phi$ direction for the $\pi^+$ dataset. Note that layers 0, 3 and 14 only have a single $\alpha$ bin each. Hence, there is no positional information to extract for these layers.

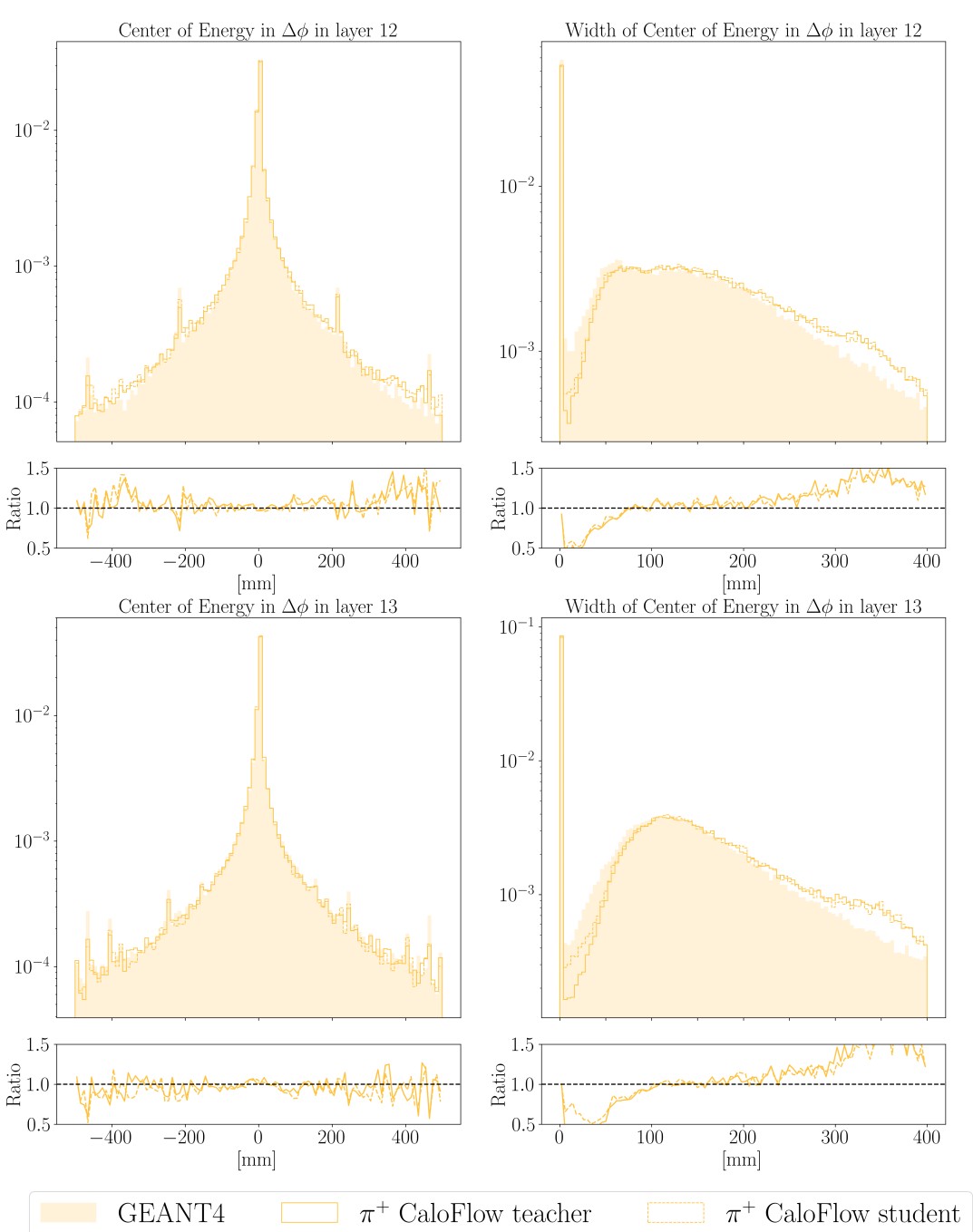

Figure 16: Distributions of the center of energy for layers 12 and 13 in the $\phi$ direction for the $\pi^+$ dataset. Note that layers 0, 3 and 14 only have a single $\alpha$ bin each. Hence, there is no positional information to extract for these layers.

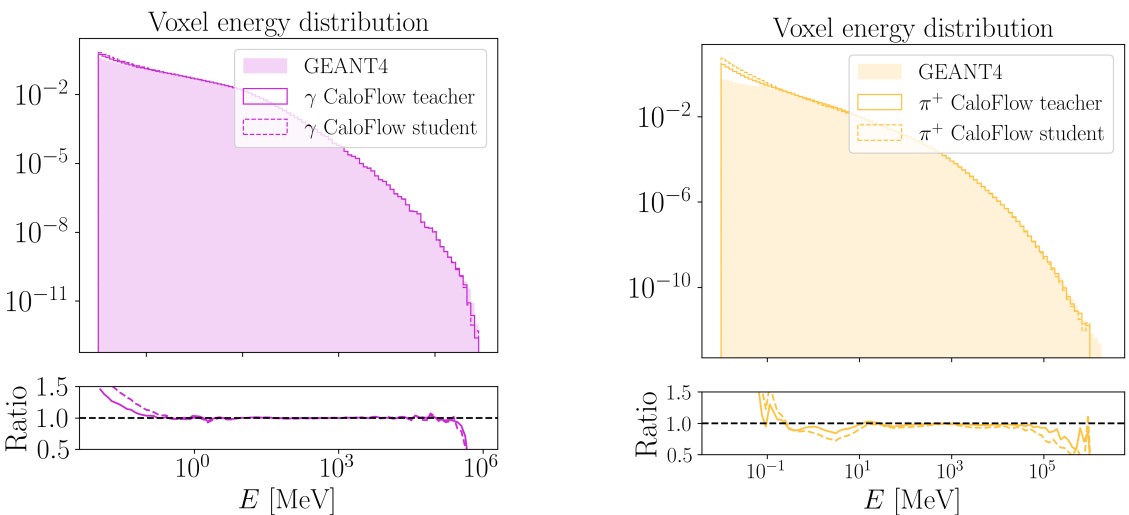

Figure 17: Distribution of voxel energies across all layers for $\gamma$ and $\pi^+$ datasets.

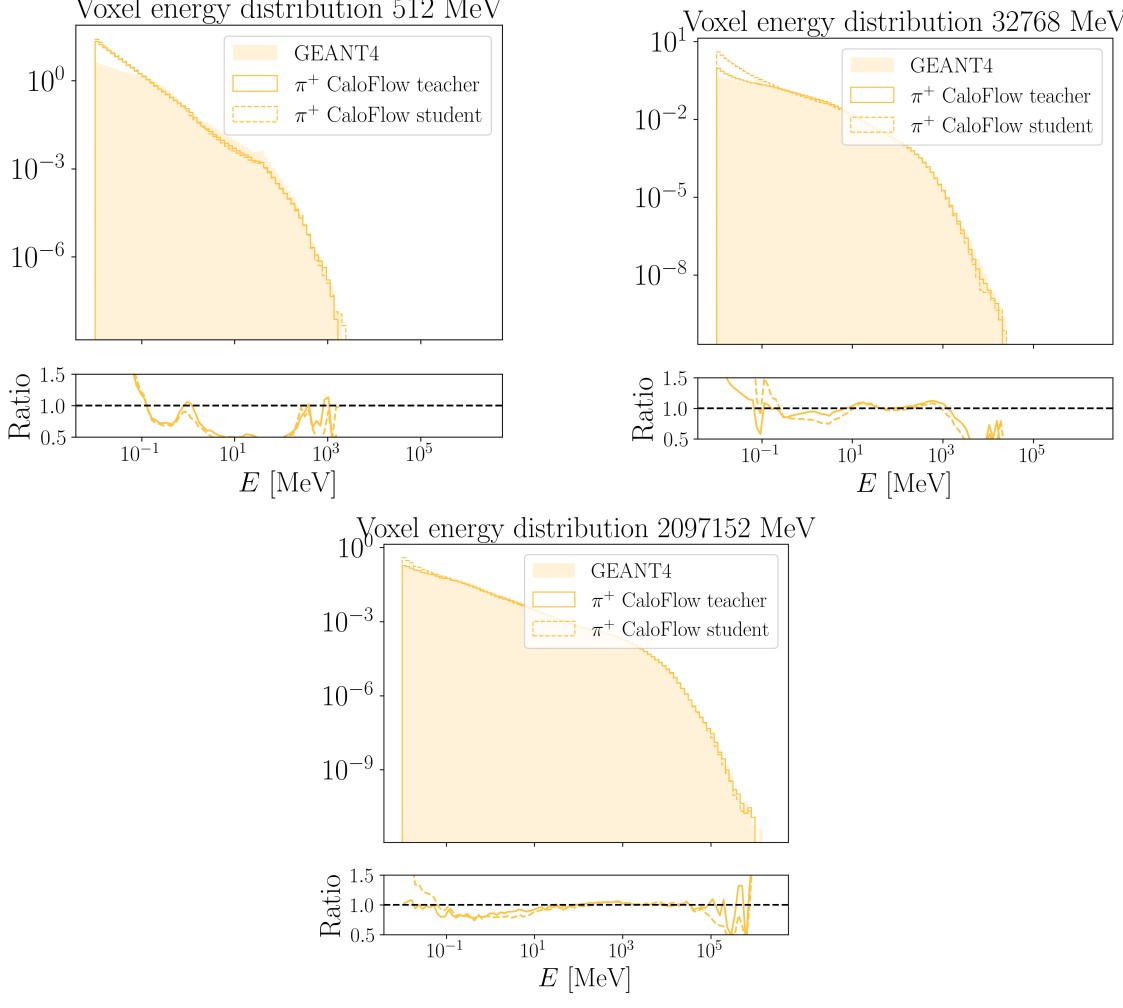

Figure 18: Distribution of voxel energies across all layers at $E_{\text{inc}} = 512$ MeV (low energy), $E_{\text{inc}} = 32768$ MeV (mid energy), and $E_{\text{inc}} = 2097152$ MeV (high energy) for $\pi^+$ dataset.

## 4.3 Classifier Scores

After looking at the similarities in 1D histograms between the CALOFLOW and GEANT4 samples in Section 4.2, it is important to perform a full phase space analysis (including correlations) by comparing the probability density that CALOFLOW learned to the one induced by GEANT4. As in [9, 10], we train a binary, neural classifier based on a fully-connected, deep neural network (DNN) to distinguish between generated CALOFLOW samples and GEANT4 samples. This serves as an approximation to the Neyman-Pearson classifier, which is the ultimate test of whether $p_{\text{generated}}(x) = p_{\text{data}}(x)$. According to the Neyman-Pearson lemma, we expect the AUC to be 0.5 if the true and generated probability densities are equal. The AUC is 1 if the classifier is able to perfectly distinguish between generated and true samples. The second metric, JSD $\in [0, 1]$, is the Jensen-Shannon divergence which also measures the similarity between the two probability distributions. The JSD is 0 if the two distributions are identical and 1 if they are disjoint.

We detail the classifier architecture and training procedure[4] in appendix B. The resulting scores are summarized in Table 5. Low-level input refers to incident energies $E_{\text{inc}}$ and the calorimeter samples in the format provided in *CaloChallenge* datasets. Training the classifier on low-level input allows us to measure the difference between the generated and reference datasets based on the voxel energies and their physical location.

Meanwhile, the "high-level" scores in Table 5 correspond to a DNN classifier trained on a set of physically-relevant high-level features, here chosen to be: incident energy $E_{\text{inc}}$, layer energies $E_i$, the center of energies and their corresponding widths in the $\eta$ and $\phi$ directions. The definitions of the center of energies and corresponding widths were stated in eqs. (4) and (5).

| AUC / JSD | | DNN based classifier | |
|---|---|---|---|
| | | GEANT4 vs. CALOFLOW (teacher) | GEANT4 vs. CALOFLOW (student) |
| | low-level (regular) | 0.701(3) / 0.092(3) | 0.739(3) / 0.131(4) |
| $\gamma$ | low-level (logit) | 0.678(4)/ 0.075(4) | 0.706(3)/ 0.100(3) |
| | high-level | 0.551(3) / 0.013(2) | 0.556(3) / 0.015(2) |
| | low-level (regular) | 0.827(3)/ 0.260(5) | 0.866(2)/ 0.341(3) |
| $\pi^+$ | low-level (logit) | 0.911(3)/ 0.457(7) | 0.914(2)/0.464(6) |
| | high-level | 0.692(2)/ 0.098(2) | 0.706(4)/ 0.108(4) |

Table 5: Classifier results, based on 10 independent runs. The first (second) numbers in every entry are AUCs (JSDs). See text for more detailed explanations.

From the scores in Table 5, we see that CALOFLOW is able to produce generated samples of sufficiently high fidelity to fool the DNN-based classifier. This is evident from most of the classifier AUC and JSD scores in Table 5 being significantly lower than unity. Overall, the teacher models performed better than the student models. Nevertheless, the student models were still able to achieve impressive classifier scores which quantitatively demonstrates the effectiveness of normalizing flows in generative modeling tasks. We note

---

[4]We also trained a classifier with the architecture and preprocessing of [12] and found quantitatively similar results to tab. 5.

that the high-level classifier scores are even better than the low-level classifier scores for a given model. This is good considering that these high-level features are probably more directly relevant for subsequent analysis steps (such as reconstruction).

Recently, the authors of CaloScore [12] have produced results for the $\gamma$ portion of Dataset 1 (as well as Datasets 2 and 3), using a score-based diffusion model. For the classifier test between CaloScore-generated and Geant4 showers (i.e. low-level features), they found an AUC of 0.98. Even accounting for the possibility of different classifier architectures and training hyperparameters, this likely indicates considerably more separability than our CaloFlow-generated showers (whose AUC is 0.739 for the low-level classifier trained on $\gamma$ showers from the student flow).

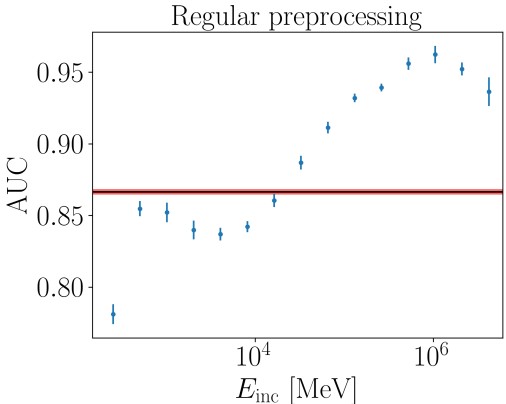 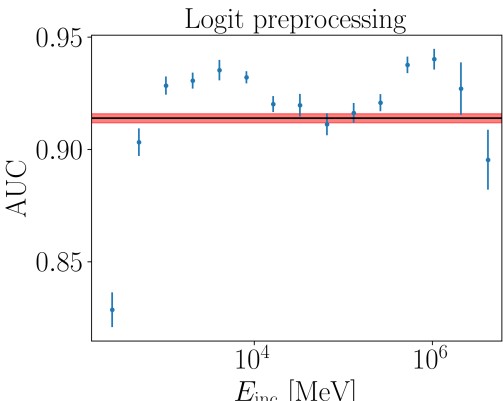

Figure 19: Plots of AUC vs $E_{\text{inc}}$. The AUCs are obtained using a classifier trained (10 independent runs) on low-level input from all showers from the CaloFlow $\pi^+$ student and Geant4 samples, and then evaluated on showers of specific $E_{\text{inc}}$. The left plot is based on voxel energy inputs with the regular proprecessing decribed in appendix B. The right plot is based on voxel energy inputs with logit preprocessing. The reference AUC based on showers of all $E_{\text{inc}}$ is shown by the horizontal line.

The low-level classifiers, with regular preprocessing as in [35], obtained from the 10 independent runs (see Table 5) were evaluated on $\pi^+$ showers with specific $E_{\text{inc}}$. The corresponding plot of AUC vs $E_{\text{inc}}$ is shown in Fig. 19 (left plot). The plot seems to imply that CaloFlow is fitting the voxel energy distribution better at low $E_{\text{inc}}$. However, this does not agree with our observations from the plots in Fig. 18. We found that preprocessing voxel energy inputs in the following way allows the classifier to become sensitive to deviations at low voxel energies: First we normalize the voxel energies by layer energies $E_i$ (such that normalized voxel energies in each layer sum to one). Then, we transform the normalized voxel energies to logit space as done during training. The overall classifier scores for this modified logit preprocessing for both particle types are also shown in Table 5.

The corresponding plot of AUC vs $E_{\text{inc}}$ for this modified preprocessing is shown in Fig. 19 (right plot). Here we see that the low $E_{\text{inc}}$ showers generally have higher AUC scores compared to the scores at the same $E_{\text{inc}}$ in the left plot. This is more consistent with our observations from Fig. 18.

## 4.4 Generation timing

We found that the average time taken to generate a single shower events by CaloFlow teacher for a generation batch size of 10000 on our TITAN V GPU is 18.91 ms for $\gamma$

| Particle type | Batch size | Student generation time per event | |
|:---:|:---:|:---:|:---:|
| | | GPU | CPU |
| $\gamma$ | 1 | 57.23 ms | 115.88 ms |
| | 10 | 5.81 ms | 12.68 ms |
| | 100 | 0.62 ms | 2.50 ms |
| | 1000 | 0.11 ms | - |
| | 10000 | **0.07 ms** | - |
| $\pi^+$ | 1 | 74.09 ms | 126.05 ms |
| | 10 | 7.46 ms | 14.03 ms |
| | 100 | 0.80 ms | 2.91 ms |
| | 1000 | 0.15 ms | - |
| | 10000 | **0.09 ms** | - |

Table 6: Average time taken to generate a single shower event by CALOFLOW student for the two particle types. The timing was computed for different generation batch sizes on our Intel i9-7900X CPU at 3.30GHz and our TITAN V GPU. We were not able to generate the shower events on the CPU for batch sizes of 1000 and 10000 due to memory constraints.

and 43.91 ms for $\pi^+$. For the same batch size, we were able to significantly reduce the generation time for CALOFLOW student.

In Table 6, we include the average time taken to generate a single shower events by CALOFLOW student for different generation batch sizes. The timings were separately evaluated on our Intel i9-7900X CPU at 3.30GHz and our TITAN V GPU. We observe that increasing the batch size reduces the generation time per shower event. However, we could not generate with large (1000 and 10000) batch sizes on the CPU due to memory constraints. We observed that the generation time is largely dependent on the sizes of layers in the MADE block (see Table 3 for MADE block layer sizes). This accounts for the difference (similarity) in generation timing between $\gamma$ teacher (student) and $\pi^+$ teacher (student). With CALOFLOW student, we are able to achieve impressively low GPU generation times of 0.07 ms per event for $\gamma$ and 0.09 ms per event for $\pi^+$.

Figure 20 compares the average CPU generation time per event found using CALOFLOW, FASTCALOGAN[5] and GEANT4. The generation times for 65 GeV and 2 TeV are shown for FASTCALOGAN and GEANT4 based on timings provided in Section 6.4 of [8]. For CALOFLOW, we included generation times for these two energies and also two other intermediate energies (262 GeV and 1 TeV). As we do not know the batch size used when timing FASTCALOGAN, we timed CALOFLOW separately using batch sizes of 1 (abbreviated as CF(1)) and 100 (abbreviated as CF(100)). We see from Figure 20 that CALOFLOW generation times are constant regardless of $E_{\text{inc}}$. In contrast, GEANT4 generation time increases with $E_{\text{inc}}$. At $E_{\text{inc}} = 65$ GeV, GEANT4 has a generation time that is $\mathcal{O}(10^2)$ slower com-

---

[5]Note that the timing of FASTCALOGAN also includes the time it takes to map the voxel energies back to calorimeter cells and we do not know how the total time splits between voxel energy generation by the GAN and cell assignment of the subsequent step.

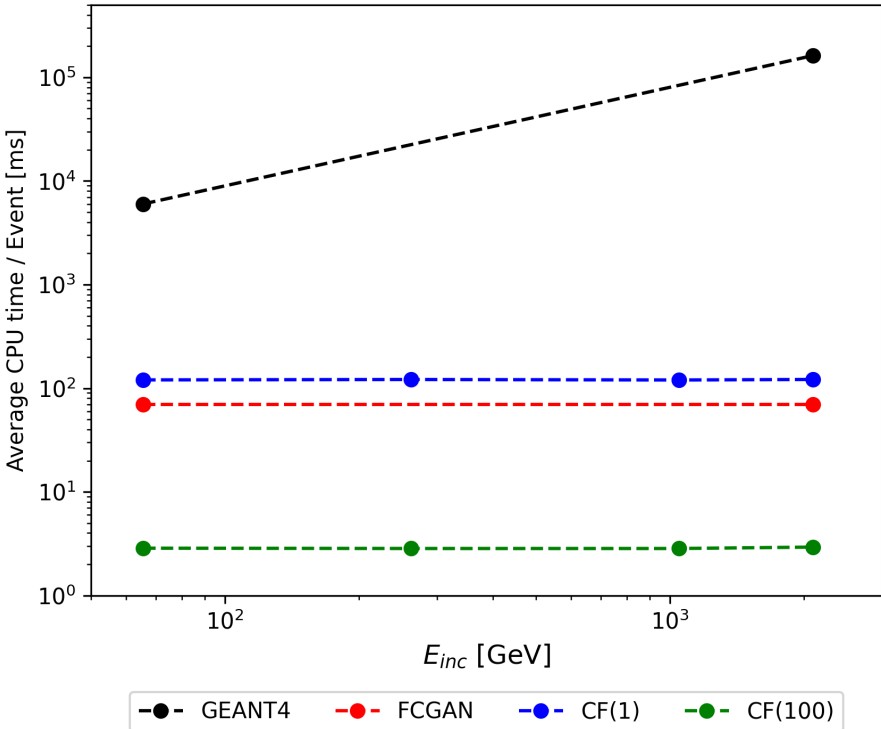

Figure 20: Comparison of CPU generation time per event for CALOFLOW (abbreviated as CF in legend), FASTCALOGAN (abbreviated as FCGAN in legend) and GEANT4 at several different $E_{\text{inc}}$. Numbers in parentheses refer to the generation batch sizes used in CALOFLOW. The CALOFLOW timing was based on our Intel i9-7900X CPU at 3.30GHz. The other timings are based on CPU used by FASTCALOGAN.

pared to CF(1) and $\mathcal{O}(10^3)$ slower compared to CF(100). At $E_{\text{inc}} = 2$ TeV, GEANT4 has a generation time that is $\mathcal{O}(10^3)$ slower compared to CF(1) and $\mathcal{O}(10^4)$ slower compared to CF(100). The timings for FASTCALOGAN are on the same order as those found using CF(1).

We can also compare directly against the results of CALOSCORE for the $\gamma$ portion of Dataset 1. Table II of [12] reports a time of 4s to generate 100 $\gamma$ showers using the score-based diffusion model approach, on a GPU with batch size of 100. Meanwhile the CALOFLOW student accomplishes the same task on similar hardware in 0.062s, which is nearly 2 orders of magnitude faster.

# 5 Conclusions/Outlook

We have shown that the CALOFLOW algorithm can be applied successfully to realistic detector simulations, namely Dataset 1 of the *Fast Calorimeter Simulation Challenge 2022*. Our results significantly outperform FASTCALOGAN, a deep generative model that was trained on the same dataset and is used in a (very limited) portion of ATLFAST3, the current fast-simulation framework of the ATLAS experiment. Based on the results shown here, we are confident that state-of-the-art deep generative models (and particularly those based on normalizing flows) will be able to play a much larger role in future versions of the fast-simulation framework.

From a machine-learning perspective, we have seen some drawbacks in the choice to use discrete incident energies, in particular concerning the need to interpolate between energies. We expect CALOFLOW to interpolate well, but with the datasets provided by the ATLAS collaboration, there is no way of checking this in detail. We think training a conditional generative model using continuous, uniformly (or log-uniformly) sampled energies, instead of discrete energies, could be advantageous and should be considered in the future.

One of the main future directions stemming from this work is figuring out how to extend the CALOFLOW framework to larger numbers of voxels, e.g. those of Datasets 2 [52] and 3 [53] of the *CaloChallenge*. Indeed, the current CALOFLOW algorithm scales badly with the number of simulated voxels and cannot be applied as-is to Datasets 2 and 3. However, we expect straightforward modifications of the basic CALOFLOW approach to be strong contenders for fast-and-accurate simulation of these higher-dimensional datasets. In the future, it will be interesting to compare flow-based approaches to Datasets 2 and 3 to other approaches, such as the score-based diffusion models of [12] (which are currently the only generative models successfully trained on Datasets 2 and 3).

It would also be interesting to consider generalizing the CALOFLOW approach beyond the incident particles considered here, and beyond the narrow $\eta$ slice. For example, it should be very straightforward to train a single conditional flow or pair of flows, conditioned on $\eta$ as well as the incident energy. This could potentially simplify the ATL-FAST3 framework, where 300 GANs were trained on individual narrow $\eta$ slices.

## Acknowledgements

We are grateful to Michele Faucci Gianelli and Ben Nachman for helpful discussions and comments on the draft. This work was supported by DOE grant DOE-SC0010008. CK would like to thank the Baden-Württemberg-Stiftung for support through the program *Internationale Spitzenforschung*, project *Uncertainties — Teaching AI its Limits* (BWST_IF2020-010). IP would like to thank the Rutgers Dept. of Physics and Astronomy for support through the Boyd scholarship.

In this work, we used the `NumPy 1.16.4` [54], `Matplotlib 3.1.0` [55], `sklearn 0.21.2` [56], `h5py 2.9.0` [57], `pytorch 1.11.0` [58], and `nflows 0.14` [59] software packages.

## A  Cyclic LR

With a cyclic LR schedule (see fig. 21a), the LR begins at a chosen base LR and then increases up to a maximum LR. The number of batches taken to increase from the base LR to the maximum LR is defined as the step size. After which, the LR decreases from the maximum LR to the base LR. This increase and subsequent decrease in LR is defined as a cycle, and this repeats for a chosen number of cycles. In certain cases, the maximum LR can be made to decrease with increasing number of batches.

OneCycle LR [49] (see fig. 21b) is a special case of cyclic LR which only relies on a single cycle followed by an annihilation phase where the base LR is gradually decreased up to a factor of 10,000.

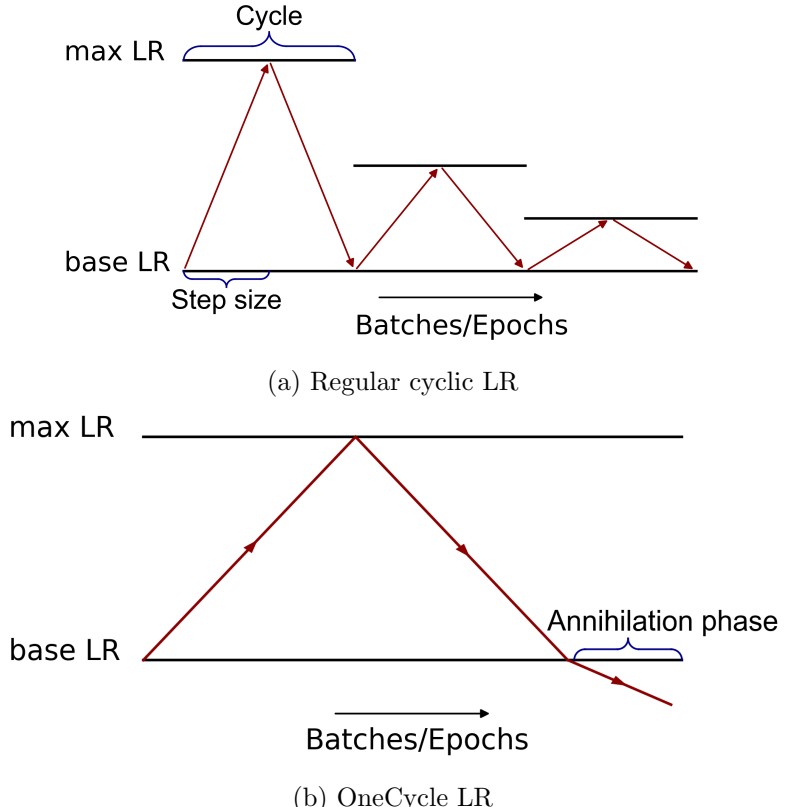

(a) Regular cyclic LR

(b) OneCycle LR

Figure 21: (a) Illustration of regular cyclic LR schedule with decreasing max LR. The length of one cycle was chosen to be 10 epochs. The step size in our study was fixed to be half the number of batches in one cycle. After each cycle, the max LR is decreased such that the difference between the max LR and the base LR is half that of the previous cycle. (b) Illustration of OneCycle LR schedule with annihilation phase.

For most of the models using OneCycle LR, the length of the annihilation phase was chosen to be 20% of the total number of training epochs shown in Table 2. The only exception was for $\pi^+$ teacher where the annihilation phase was chosen to be 10% of the total number of training epochs. The step size was taken to be half of the remaining number of epochs.

For $\gamma$ teacher, which used regular cyclic LR, the length of one cycle was chosen to be 10 epochs. After each cycle, the max LR is decreased such that the difference between the max LR and the base LR is half that of the previous cycle. The step size in our study was fixed to be 5 epochs (i.e. half of a cycle).

The base and max LRs used when training the various models are shown in Table 7.

## B  Classifier Architecture

The classifier-based performance metric uses the classifier architecture that was provided in the evaluation script of the *CaloChallenge* [35]. It is based on the classifiers that were used in [9, 10]. In detail, the classifier is a deep, fully-connected neural network with an input and two hidden layers with 512 nodes each. The output layer returns a single number which is passed through a sigmoid activation function. All other activation functions are leaky ReLUs, with default negative slope of 0.01. We do not use any dropout or batch

|   |         | base LR | max LR |
|---|---------|---------|--------|
| $\gamma$ | Teacher | $5 \times 10^{-5}$ | $2 \times 10^{-3}$ |
|   | Student | $4 \times 10^{-5}$ | $1 \times 10^{-3}$ |
| $\pi^+$ | Teacher | $4 \times 10^{-5}$ | $1 \times 10^{-3}$ |
|   | Student | $2 \times 10^{-5}$ | $1 \times 10^{-3}$ |

Table 7: Base and max LRs used when training $\gamma$ and $\pi^+$ teacher/student models. For $\gamma$ teacher, the max LR here refers to the max LR in the first cycle.

normalization regulators.

In the classifier of the low-level features, we use as input the incident energy (preprocessed as $\log_{10} E_{\mathrm{inc}}$) and the energy deposition in each voxel (preprocessed as $\mathcal{I}/E_{\mathrm{inc}}$). High-level features are the incident energy (preprocessed as $\log_{10} E_{\mathrm{inc}}$), the energy deposited in each layer (preprocessed as $\log_{10}(E_i + 10^{-8})$), the center of energy in $\eta$ (normalized with a factor 100), the center of energy in $\phi$ (normalized with a factor 100), the width of the $\eta$ distribution (normalized with a factor 100), and the width of the $\phi$ distribution (normalized with a factor 100).

We split all the data in train/test/validation sets of the ratio (60:20:20). For the both the $\gamma$ and $\pi^+$ showers, we used the second file that was provided at [36]. From our flow models, we used a generated sample of the same size and $E_{\mathrm{inc}}$ distribution as the GEANT4 data.

The networks are then optimized by training 50 epochs with an ADAM [60] optimizer with initial learning rate of $2 \cdot 10^{-4}$ and a batch size of 1000, minimizing the binary cross entropy. We use the model state with the highest accuracy on the validation set for the final evaluation and we subsequently calibrate the classifier using isotonic regression [61] of `sklearn` [56] based on the validation dataset before evaluating the test set.

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
