# Peer review of "CaloFlow for CaloChallenge Dataset 1"

_SciPost Physics_

## Round 2 · Referee Report · Anonymous (Referee 1) · 2024-1-8

Strengths

  1. Rigorous report on a study relevant for collider physics applications
  2. High quality of presentation

Weaknesses

  1. Incremental work on application of an existing tool to a new dataset
  2. Generality performance relevant to realistic production applications unclear
  3. Comparison to current leading tool not performed on the exact same dataset: would be stronger as a detailed apples-to-apples comparison with attention to correlations

Report

This is a solid piece of work on application of the CaloFlow ML architecture to a calorimeter fast-sim data challenge. The work is well-presented and rigorous, though there are some hanging questions of generalisability to continuous energy distributions, and in the extent to which strong conclusions can be drawn from comparison to an alternative model with a related but not identical dataset. There is for me a question of the level of interest in the material: this feels more like something to be part of a report on the CaloChallenge than an independent paper. Indeed, the CaloFlow method is already established, meaning that this standalone performance on a different dataset is of marginal or technical interest: the bigger question would be a head-to-head detailed comparison of different tools' performances on the same dataset, which would need to be a collaborative publication between the various teams. Comparing to the publication requirements for SciPost Physics I find that this does not have the breakthrough or novelty required, and is more appropriate for Physics Core, as a more incremental study-update.

Requested changes

  1. p2 "Dataset" section: please give the reason for using only Dataset 1 when the other two are available. Is the motivation that the target for comparison is AtlFast3, and this is its training set? Is it possible to achieve significantly better performance, e.g. in some of the misdescribed regions, with the higher-dimensional datasets?

  2. p3: several differences in setup (number of voxels, energy distributions) with respect to CaloGAN are noted: why are these differences present, and do they impact on the conclusions about relative performance of the approaches? If the aim is primarily to compare to FastCaloGAN, why use a qualitatively different dataset?

  3. p3: in particular, why discretely sample energies rather than smoothly sample? Does this impact the generality, e.g. is the behaviour smoothly interpolated for energies between the powers of two used in the training set?

  4. p6 3a: how were these sometimes very precise layer sizes decided? Is the behaviour with different layer sizes really as hit and miss as the text suggests? It is concerning that the question of appropriate architecture seems so unpredictable.

  5. p9 4.1: what slight difference between pi+ layer-0 images do you wish to highlight?

  6. p10, 11: please define these average shower images: is there any angular standardisation involved or are they just naively averaged and hence rotationally symmetric? (In which case a set of radial histograms would be clearer for comparisons.) Are they averaged over all the discrete energies? And how to understand the physics of the images: is it possible to infer what each layer is doing, e.g. with regard to the annular band structures, or to understand the physics of the combination of layers in expressing photon and pion shower development?

  7. p11 4.2.1: what are the origins of the spikes at high energy in the E2 histogram? This looks like a spurious behaviour for this particular energy, and a naive interpretation would be a lack of statistics. Is this a case of GIGO?

  8. p13 bottom and p14 mid: a reference is made to worse performance in an AtlFast3 paper: it would help the reader to actually reproduce the plots here for direct comparison, rather than requiring a manual cross-referencing.

  9. p14 bottom: is there any identifiable reason for this significantly improved performance? E.g. a reason for expecting flows to be a better-suited generative model? Or is the difference due to the different voxel numbers, preprocessing, or discrete energies wrt the CaloGAN training? These differences make it hard to conclude that the two tools differ in performance on the exact same application.

  10. As a general comment, it would be nice to see some diagnostics looking at the degrees of correlation between different observables: it would be possible to perform a "smearing"-type sampling from these 1D distributions and achieve (by construction) a reasonable performance, while the potential of ML approaches is to provide shower description with full correlations in a higher-dimensional observable space.

  11. p27 Sec 5: application of this model in production would require integration with the C++ simulation frameworks: is this currently practical? I believe flow models (or at least some frameworks' flow models) can't currently be fully preserved and communicated via the ONNX format; are there C++ flow frameworks, and ways to communicate the Python-trained architectures into them? Does the deployment require GPU-enabled production nodes, or can it be run in CPU mode? As the majority of current grid resources are CPU-based, what would the effect of CPU-based CaloFlow running be on the timing performance wrt (also CPU-based) Geant4?

  • validity: good
  • significance: ok
  • originality: low
  • clarity: high
  • formatting: perfect
  • grammar: excellent

Author:  Claudius Krause  on 2024-04-10  [id 4406]

(in reply to Report 1 on 2024-01-08)

Dear referee,

Thank you very much for your detailed comments!

We want to point out that the Challenge dataset 1 is substantially different from the CaloGAN dataset. - The calorimeter considered here is the official ATLAS ECal+HCal. As such, it is a sampling calorimeter with a sampling fraction < 1. The original CaloGAN dataset on which CaloFlow was developed had a sampling fraction of 1 (apart from small leakage), which lead to some particular design choices. We show how these translate to the new setup. - The training data here was chosen by ATLAS to be discrete and increasing in factors of 2. This is different to what CaloFlow was trained on (uniformly distributed incident energies), and we showed that this change does not affect the quality of the training.
- The voxelization here contains more layers, which increases the complexity of the dataset. It was a priori not clear if the algorithm will handle such a case as well as before. - The voxelization here was done in cylindrical coordinates, so when represented as 2d arrays, some neighboring voxels are not in adjacent array positions. This poses a further complication of the dataset which we showed to be well modeled by CaloFlow. We do compare to the available model back then, FastCaloGAN. A more detailed comparison of all available methods will indeed be part of the CaloChallenge write-up. We therefore believe to have presented substantial new material and be publishable in SciPost Physics instead of Physics Core.

1. p2 "Dataset" section: please give the reason for using only Dataset 1 when the other two are available. Is the motivation that the target for comparison is AtlFast3, and this is its training set? Is it possible to achieve significantly better performance, e.g. in some of the misdescribed regions, with the higher-dimensional datasets?

Datasets 2 and 3 have no overlap with dataset 1 (different detector setup and Geant4 simulation), so working on datasets 2 and 3 would have no impact on the performance on dataset 1. As to why we only target dataset 1 here, this is because of the dimensionality of the problems. Datasets 2 and 3 have a 10 and 100 times larger number of voxels, making the application of the ‘plain’ Caloflow algorithm computationally (nearly) impossible. We targeted those with a separate algorithm in a separate publication. We added a sentence to that paragraph explaining the reason.

** 2. p3: several differences in setup (number of voxels, energy distributions) with respect to CaloGAN are noted: why are these differences present, and do they impact on the conclusions about relative performance of the approaches? If the aim is primarily to compare to FastCaloGAN, why use a qualitatively different dataset?**

CaloGAN was a toy calorimeter configuration loosely based on the ATLAS ECAL. It had a number of major simplifications including the assumption that the sampling fraction was 100% (essentially that there were no inactive layers). Meanwhile, CaloChallenge Dataset 1 is from the actual, official, fully realistic ATLAS GEANT4 detector simulation of both ECAL and HCAL. The goal of this work was to extend the approach of CaloFlow, which was successfully demonstrated on the CaloGAN dataset, to the more complex and realistic setting of CaloChallenge Dataset 1. We present the differences between the CaloGAN data and dataset 1 here, to explain which parts of CaloFlow needed to be adapted to the new dataset. The terminology is confusing, but FastCaloGAN was developed by ATLAS for this latter dataset, not the CaloGAN dataset. So it is the appropriate point of comparison for our work. Since dataset 1 of the CaloChallenge was provided by the ATLAS collaboration, we have to take it as is and cannot perform additional studies, for example on the difference between discrete and continuous incident energies. These questions have been considered inside ATLAS in the development of AtlFast 3 and FastCaloGAN.

3. p3: in particular, why discretely sample energies rather than smoothly sample? Does this impact the generality, e.g. is the behaviour smoothly interpolated for energies between the powers of two used in the training set?

Because the dataset provided by the Challenge (and used by FastCaloGAN and AtlFast3) was chosen to be that way. As already mentioned above, this was not our choice. ATLAS claimed interpolation works well, but without additional data, which would need to come from ATLAS, we cannot tell.

4. p6 3a: how were these sometimes very precise layer sizes decided? Is the behaviour with different layer sizes really as hit and miss as the text suggests? It is concerning that the question of appropriate architecture seems so unpredictable.

While the presented numbers indeed seem unpredictable, we started with layer sizes 1x or 2x times the number of voxels and iterated from there. We did not perform a detailed multidimensional study of hyperparameters.

5. p9 4.1: what slight difference between pi+ layer-0 images do you wish to highlight?

There is a slight difference in the average energies in one of the radial bins. But since that is very small, we decided to remove that sentence to avoid confusion to the reader.

6. p10, 11: please define these average shower images: is there any angular standardisation involved or are they just naively averaged and hence rotationally symmetric? (In which case a set of radial histograms would be clearer for comparisons.) Are they averaged over all the discrete energies? And how to understand the physics of the images: is it possible to infer what each layer is doing, e.g. with regard to the annular band structures, or to understand the physics of the combination of layers in expressing photon and pion shower development?

They are naively averaged across all incident energies, but not rotationally symmetric, because the data is not from the central eta slice. These figures are meant to illustrate how well the model covers the entire phase space of the detector. (The original GAN showers of the CaloGAN paper already had noticeable differences at that level when compared to Geant4 and the original CaloFlow improved that a lot. For this historic reason, we decided to include these figures here, too).

7. p11 4.2.1: what are the origins of the spikes at high energy in the E2 histogram? This looks like a spurious behaviour for this particular energy, and a naive interpretation would be a lack of statistics. Is this a case of GIGO?

These spikes come from the discrete incident energies, which increase by factors of 2 and therefore have substantially more energy depositions from one energy to the next. They are not an artifact of small statistics, but they are present in the training data and well reproduced by our model.

8. p13 bottom and p14 mid: a reference is made to worse performance in an AtlFast3 paper: it would help the reader to actually reproduce the plots here for direct comparison, rather than requiring a manual cross-referencing.

We digitized the ATLAS plots, since their samples are not public, and added it to our figures.

9. p14 bottom: is there any identifiable reason for this significantly improved performance? E.g. a reason for expecting flows to be a better-suited generative model? Or is the difference due to the different voxel numbers, preprocessing, or discrete energies wrt the CaloGAN training? These differences make it hard to conclude that the two tools differ in performance on the exact same application.

We again apologize for the confusing terminology (which we have no control over). As noted above in the reply to question 2, the comparison in this work with AtlFast3/FastCaloGAN is fully “apples-to-apples” in that they are being trained on the same dataset (“official” ATLAS GEANT4 ECAL+HCAL simulation = CaloChallenge Dataset 1). The observed improvement in performance is therefore entirely on Normalizing Flows being a superior architecture compared to GANs. We have discussed this extensively in the CaloFlow papers, and many other works in the literature by now. Normalizing Flows are trained with the maximum likelihood objective instead of the min-max objective, which leads to much more stable, convergent training, less issues with mode collapse, and principled model selection. We added a short sentence to the paragraph. In the CaloFlow works, we performed a similar apples-to-apples comparison with CaloGAN on the CaloGAN toy dataset and demonstrated a similarly huge improvement in performance. We are seeing the same thing here with the CaloChallenge Dataset 1 relative to AtlFast3/FastCaloGAN.

10. As a general comment, it would be nice to see some diagnostics looking at the degrees of correlation between different observables: it would be possible to perform a "smearing"-type sampling from these 1D distributions and achieve (by construction) a reasonable performance, while the potential of ML approaches is to provide shower description with full correlations in a higher-dimensional observable space.

The classifier test of Section 4.3 aims precisely at this question. By learning to distinguish generated showers from Geant showers based either on all voxels (low-level observables), or the observables shown in the histograms (high-level observables), all correlations between these variables are taken into account. We added a note that correlations are included in this test into Section 4.3.

11. p27 Sec 5: application of this model in production would require integration with the C++ simulation frameworks: is this currently practical? I believe flow models (or at least some frameworks' flow models) can't currently be fully preserved and communicated via the ONNX format; are there C++ flow frameworks, and ways to communicate the Python-trained architectures into them? Does the deployment require GPU-enabled production nodes, or can it be run in CPU mode? As the majority of current grid resources are CPU-based, what would the effect of CPU-based CaloFlow running be on the timing performance wrt (also CPU-based) Geant4?

This algorithm runs on CPUs too, but it is slower, as can be seen in the timing comparison in Table 7. An integration of such a model into C++ frameworks via ONNX or other interfaces should be possible, but is a software development problem that we will leave for future work. In addition, whether or not this algorithm will be implemented also depends on the final evaluation of the CaloChallenge, which also includes timing and other resource studies of all submissions on comparable hardware.

---

## Round 2 · Referee Report · Tilman Plehn (Referee 2) · 2024-2-8

Report

I have to admit that I am slightly confused about the time evolution of this preprint. It is really from the Fall of 2022? For the report this is a serious problem, because in some subfields of LHC physics a year actually makes a difference...

Essentially, I agree with many points of the first referee, but I assume a lot them are by now sorted out in other papers or will be sorted out in the actual CaloChallenge draft?

This paper is something like a classic by now, one of the key papers for the CaloChallenge with results presented at ML4Jets 2023. On the arXiv it has 23 citations in a good year, which means it enjoys serious visibility. That means it should definitely be published, even though some of the results might by now not be state of the art anymore.

As the authors know, one of my standard complaints is that the citation culture between generative networks for detector simulations and the same networks for event generation is not helping our young people. If the authors feel like it, I would appreciate some kind of virtual time travel and a paragraph on similar networks applied to other aspects of LHC physics.

Concerning Sec. 3.2, it is weird, because by now we all know how this architecture could be improved in expressivity and in speed. Publishing it as is in SciPost might be confusing to readers, maybe a short footnote could point out that the paper really was technically the state of the art when it was submitted to the arXiv? Surely, we cannot hold the authors reponsible for the delay. Still, do we need Fig.3 to illustrate OneCycle as a technical detail?

In Sec. 4 the authors describe their dataset, which is definitely still valid, and those figures have been used as an inspiration and an illustration of many papers since. For the future, what about adding secondary panels with bin-wise ratios of the different curves? The description of the dataset, the underlying physics, and the challenges it poses to ML-generators are still valid and warrant publication.

Finally, I would very much appreciate it if everyone could avoid bringing me into this situation in the future. I know that writing this report took me more than a months, which is on the long side. But you could have asked me right after the paper was submitted, and I would have been able to provide more useful feedback then. At this stage I can only recommend publishing this paper asap, because all improvements I can come up with are already published on the arXiv, so what's the point of proposing them now?
  • validity: -
  • significance: -
  • originality: -
  • clarity: -
  • formatting: -
  • grammar: -

Author:  Claudius Krause  on 2024-04-10  [id 4407]

(in reply to Report 2 by Tilman Plehn on 2024-02-08)

Dear Tilman,

Thank you very much for your detailed comments! Below, you find our replies.

**I have to admit that I am slightly confused about the time evolution of this preprint. It is really from the Fall of 2022? For the report this is a serious problem, because in some subfields of LHC physics a year actually makes a difference…**

We apologize for the part of the delay that is on us, but we have no control over the process at SciPost.

**Essentially, I agree with many points of the first referee, but I assume a lot them are by now sorted out in other papers or will be sorted out in the actual CaloChallenge draft?**

Yes, a detailed comparison of quality and resource requirements of the different models will be part of the CaloChallenge evaluation that is currently being prepared.

**This paper is something like a classic by now, one of the key papers for the CaloChallenge with results presented at ML4Jets 2023. On the arXiv it has 23 citations in a good year, which means it enjoys serious visibility. That means it should definitely be published, even though some of the results might by now not be state of the art anymore.**

Thank you very much for this positive feedback. We hope to have presented enough new material (see our reply to the very first point by reviewer 1) to be considered for SciPost Physics instead of SciPost Physics Core.

**As the authors know, one of my standard complaints is that the citation culture between generative networks for detector simulations and the same networks for event generation is not helping our young people. If the authors feel like it, I would appreciate some kind of virtual time travel and a paragraph on similar networks applied to other aspects of LHC physics.**

Yes, we agree. We added more references to the introduction.

**Concerning Sec. 3.2, it is weird, because by now we all know how this architecture could be improved in expressivity and in speed. Publishing it as is in SciPost might be confusing to readers, maybe a short footnote could point out that the paper really was technically the state of the art when it was submitted to the arXiv? Surely, we cannot hold the authors reponsible for the delay. Still, do we need Fig.3 to illustrate OneCycle as a technical detail?**

We moved the one cycle learning rate schedule to an appendix.

**In Sec. 4 the authors describe their dataset, which is definitely still valid, and those figures have been used as an inspiration and an illustration of many papers since. For the future, what about adding secondary panels with bin-wise ratios of the different curves? The description of the dataset, the underlying physics, and the challenges it poses to ML-generators are still valid and warrant publication.**

Thank you for this suggestion, we added the bin-wise ratios to the figures in Section 4.

---

## Editorial Decision

resubmitted